# Telomerase RNA structural heterogeneity in living human cells detected by DMS-MaPseq

Nicholas M. Forino [1,7], Jia Zheng Woo[2,7], Arthur J. Zaug[3,4], Arcelia Gonzalez Jimenez[5], Eva Edelson [6], Thomas R. Cech[3,4] ✉, Silvi Rouskin [2] ✉ & Michael D. Stone [5] ✉

Biogenesis of human telomerase requires its RNA subunit (hTR) to fold into a multi-domain architecture that includes the template-pseudoknot (t/PK) and the three-way junction (CR4/5). These hTR domains bind the telomerase reverse transcriptase (hTERT) protein and are essential for telomerase activity. Here, we probe hTR structure in living cells using dimethyl sulfate mutational profiling with sequencing (DMS-MaPseq) and ensemble deconvolution analysis. Approximately 15% of the steady state population of hTR has a CR4/5 conformation lacking features required for hTERT binding. The proportion of hTR CR4/5 folded into the primary functional conformation is independent of hTERT expression levels. Mutations that stabilize the alternative CR4/5 conformation are detrimental to telomerase assembly and activity. Moreover, the alternative CR4/5 conformation is not found in purified telomerase RNP complexes, supporting the hypothesis that only the primary CR4/5 conformer is active. We propose that this misfolded portion of the cellular hTR pool is either slowly refolded or degraded, suggesting that kinetic RNA folding traps studied in vitro may also hinder ribonucleoprotein assembly in vivo.

Telomeres are repetitive regions of DNA found at chromosome ends that are sheathed with a coat of protective proteins[1,2]. Telomerase is a ribonucleoprotein (RNP) enzyme that catalyzes telomere DNA repeat synthesis using a short segment of its integral RNA subunit as a template[3]. The catalytic action of telomerase counteracts the gradual erosion of telomeric DNA arising from incomplete replication of the lagging strand by the DNA replication machinery[4,5]. In the absence of telomerase, telomeres eventually shorten to critical lengths that can trigger cellular senescence or programmed cell death[6,7]. Telomere length maintenance by telomerase is thus an essential process for ensuring the genomic stability and replicative capacity of rapidly dividing or continually regenerating tissues[8,9].

Underpinning the action of telomerase is the multistep co-assembly of hTR, hTERT, and several additional proteins into the functional RNP complex (Fig. 1a)[10,11]. Human telomerase is structurally organized by the hTR subunit, which adopts a phylogenetically conserved multi-domain architecture and serves as the binding scaffold for telomerase proteins[12,13] (Fig. 1b). Prior chemical probing studies of hTR[14,15] were instrumental in providing direct biochemical evidence of the proposed phylogenetically conserved secondary structure[12]. However, these studies came to differing conclusions on whether hTERT binds a pre-organized hTR or induces hTR remodeling. Central to this discrepancy is that these studies were technically limited to gathering structural information on the population average of what has been shown to be a heterogeneous pool of hTR reservoirs[16,17]. Overexpression of telomerase has also historically been used to mitigate problems arising from its scarcity but has been shown to bypass the endogenous RNP assembly pathway[18].

[1]Department of Molecular, Cell, and Developmental Biology, University of California, Santa Cruz, CA, USA. [2]Department of Microbiology, Harvard Medical School, Boston, MA, USA. [3]Department of Biochemistry, University of Colorado, Boulder, CO, USA. [4]Howard Hughes Medical Institute, University of Colorado, Boulder, CO, USA. [5]Department of Chemistry and Biochemistry, University of California, Santa Cruz, CA, USA. [6]Department of Microbiology and Environmental Toxicology, University of California, Santa Cruz, CA, USA. [7]These authors contributed equally: Nicholas M. Forino, Jia Zheng Woo. ✉e-mail: cech@hhmi.org; silvi@hms.harvard.edu; mds@ucsc.edu

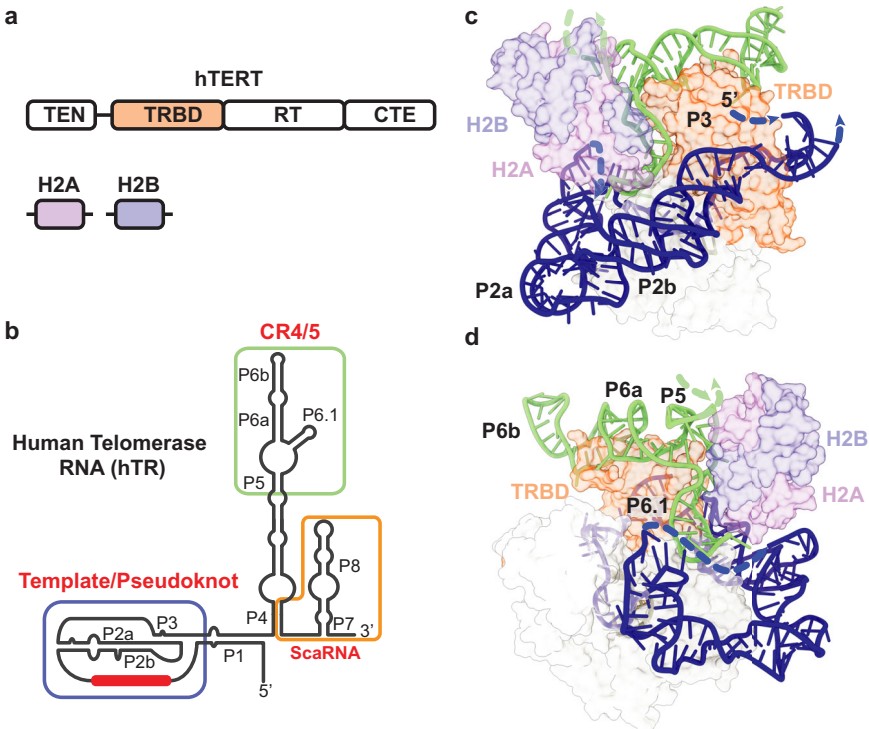

**Fig. 1 | Overview of human telomerase components and hTR structure. a** The conserved domain organization of TERT includes the telomerase essential amino-terminal domain (TEN), RNA binding domain (TRBD), reverse transcriptase domain (RT), and carboxy-terminal extension (CTE). A H2A/B dimer comprises one H2A and one H2B protein. **b** The vertebrate-conserved architecture of hTR includes the template/pseudoknot (t/PK, template sequence shown as red bar), CR4/5, and scaRNA domains. P, paired region. **c** The hTR template/pseudoknot domain (in blue) wraps around TERT and forms the pseudoknot helix P3. TRBD, telomerase RNA binding domain. **d** CR4/5 (in green) adopts an 'L' shaped three-way junction and sandwiches TERT TRBD between the P6.1 and P6a/b stems. A dimer of H2A/B binds P5 and P6.1. Structure figures made using PDB 7BG9.

Recent cryogenic electron microscopy (cryo-EM) discoveries have illuminated the structure and molecular composition of the *Tetrahymena* and human telomerase enzymes[19–25]. While both enzymes share important structural features in the catalytic domain, human telomerase has a uniquely bilobed architecture[10,11,26]. The catalytic lobe contains the hTERT protein, the conserved RNA t/PK and CR4/5 domains, and an hTR-bound histone dimer (Fig. 1c, d). The biogenesis lobe includes two complements of a core tetramer of H/ACA proteins bound to a small Cajal-body RNA domain (scaRNA) near the 3' end of hTR[27,28]. Assembly of telomerase involves a multistep cascade of protein-binding events and nuclear trafficking of the immature telomerase RNP before it finally arrives at telomeres as an active complex[29,30]. After transcription, assembly of the H/ACA RNP licenses hTR for maturation and prevents its degradation by the RNA exosome[31,32]. hTR transits through the nucleolar and Cajal body sub-nuclear compartments and accumulates in the nucleoplasm. Association with hTERT excludes hTR from the nucleolus while association with the TCAB1 permits transient association with Cajal bodies[33–35]. While the precise spatio-temporal details of hTERT association with immature telomerase RNPs remain unclear, it is well-established that mutations in telomerase components that disrupt RNP biogenesis can cause a wide range of inherited diseases[36,37]. Moreover, mutagenesis studies on hTR have reinforced the importance of the t/PK and CR4/5 adopting precise conformations to achieve the assembly of a catalytically active telomerase RNP[38–40].

Here, we identify and characterize alternative conformations of endogenous hTR in living cells through a targeted DMS-MaPseq approach coupled with ensemble deconvolution[41,42]. Across the entire population of hTR molecules, DMS-reactive nucleotides are encoded as mutations in sequencing reads. Thus, each read represents a single-molecule measurement of hTR structure, permitting the use of bioinformatic clustering to identify groups of reads that are defined by distinct mutation patterns. We used DMS constraints derived from the clustered MaPseq results to guide RNA folding predictions, revealing that the t/PK and CR4/5 domains each exist in a structural ensemble that includes their canonically described conformation and an alternative conformation. Based on current cryo-EM models of the assembled telomerase RNP[19,24], these alternative t/PK and CR4/5 conformations are not representative of hTR in its hTERT-bound form. In fact, we found that alternatively folded hTR conformers persist when hTERT is absent or overexpressed, suggesting an RNA folding activity of other hTR-interacting factors. CR4/5 mutants designed to destabilize the canonical three-way junction conformation while stabilizing the alternative conformation to varying degrees were misfolded in cells and exhibited catalytic defects. The decrease in activity is explained by impaired RNP assembly. Taken together, our results suggest that the in vivo hTR folding pathway is complex and includes alternative conformations that may represent dead-end misfolded states or possible intermediates along the telomerase RNP biogenesis pathway. In either case, our results suggest that assembly of catalytically active telomerase RNPs may be limited by a non-productive conformation of the CR4/5 domain.

## Results

### DMS-MaPseq reveals discrepancies with the established hTR structure model

To test whether our MaPseq approach could capture structural features of endogenous hTR in living cells, we first generated and analyzed DMS reactivities of the ensemble average in HeLa cells. Visual comparison of measured DMS reactivities of the t/PK domain

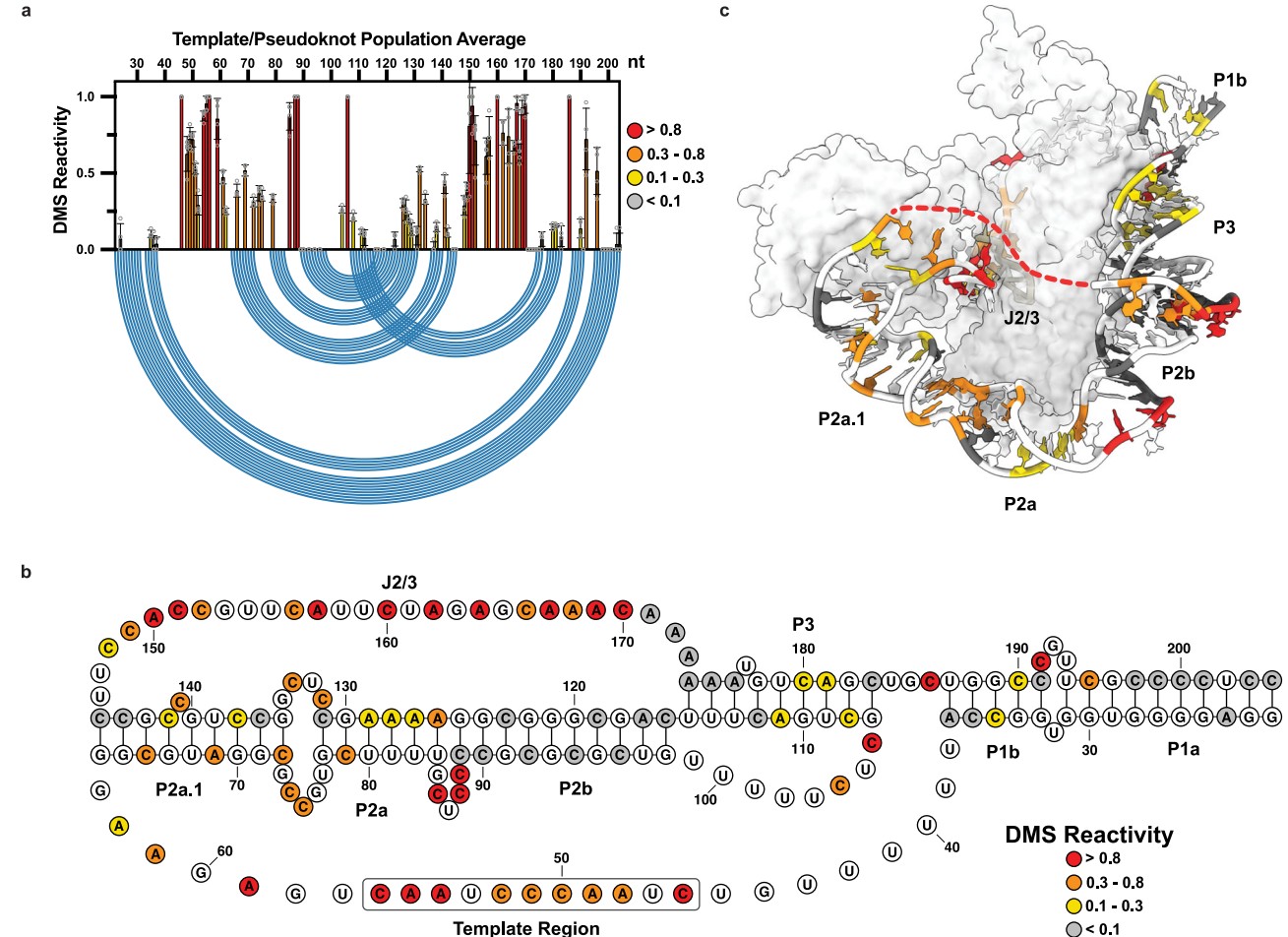

**Fig. 2 | Population average DMS reactivity of the hTR t/PK domain.**
**a** Normalized DMS reactivity of hTR t/PK domain (nt. 22–204, $n = 4$ biological replicates). Intensity of DMS reactivity colored according to the provided legend. Data are presented as means ± SD, open circles representing individual replicate values. Blue arcs designate the base pairing pattern of the canonical t/PK conformation seen by cryo-EM of assembled telomerase. **b** Secondary structure of the canonical t/PK conformation with DMS reactivity of the hTR population average overlaid onto the nucleotides. Stem elements (P), joining regions (J) and the template sequence are labeled. **c** DMS reactivity overlaid onto the cryo-EM model of assembled telomerase. PDB 7BG9. Source data are provided as a Source Data file.

(Fig. 2a, b) with recent cryo-EM derived models of assembled telomerase revealed many regions of good agreement[19] (Fig. 2c). For example, we observed nucleotides with medium to strong degrees of DMS reactivity (0.3–0.8 and >0.8, respectively) in the template region (C46-C56), the internal loop joining the P2a.1 and P2a stems, the bulge joining P2a and P2b (J2a/b), and hinge region linking P2a.1 to P3 (J2a.1/3), indicative of single-stranded RNA. High DMS reactivity of template nucleotides in vivo has also been observed for *Tetrahymena* telomerase[43]. DMS reactivity of the CR4/5 domain was also largely consistent with a three-way junction architecture, with most of the reactive nucleotides residing within the internal loops that join the P4.2/P5 stems and the P6a/P6b stems, the apical loop of P6 (C277-278), the junction regions of the three-way junction motif (C247, C248, A252, C255, A301, C317, A318), and nucleotides comprising the P6.1 stem (A302, A304, A311, A313) (Fig. 3a–c).

To more quantitatively evaluate agreement of our DMS reactivity data with widely accepted hTR secondary structure models reported by Chen et al.[12], we employed the area under the receiver operating characteristic curve (AUROC) (Fig. 4). The ROC curve depicts the balance between precision, represented by the true positive rate, and sensitivity, indicated by the false positive rate, when classifying data into two distinct categories. In the context of treating an RNA secondary structure model as a binary classifier, the ROC curve

differentiates between paired and unpaired bases at varying thresholds of DMS signal intensity. At each threshold, it computes the true positive rate and false positive rate in comparison to the provided RNA structural model. An AUROC score of 1.0 signifies the existence of a specific threshold where all DMS-reactive positions correctly align with unstructured regions and all DMS-unreactive positions are found in regions of RNA base pairing. We note that DMS modification relies upon the solvent accessibility of the N1 of adenine and N3 of cytosine – a factor not accounted for in secondary structure models. Therefore, achieving the perfect AUROC for a given secondary structure model may not be feasible due to the influence of the underlying three-dimensional structure of the target RNA. Previous work has established that AUROC values of ~0.9 or higher indicate a high degree of concordance between the structural model and DMS modification pattern, whereas AUROC values approaching ~0.5 arise in the case of little to no agreement between the structural model and probing data[44]. Across four independent biological replicates, the AUROC values for the DMS reactivity when evaluated against the Chen model for the t/PK and CR4/5 domains were 0.93 ± 0.003 and 0.91 ± 0.01, respectively (Fig. 4a). These AUROC values demonstrate good agreement between widely accepted hTR secondary structure models for the t/PK and CR4/5 domains and the DMS reactivity patterns obtained by our in vivo probing experiments.

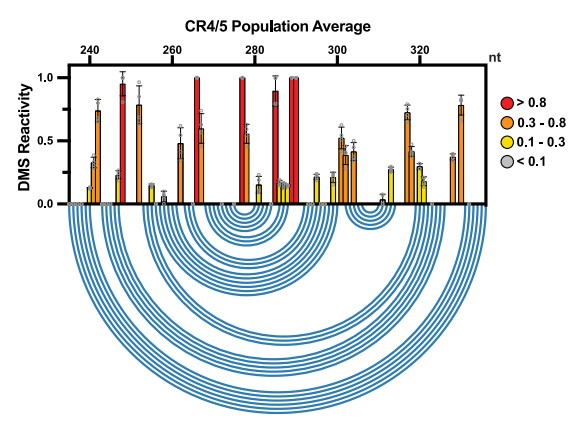

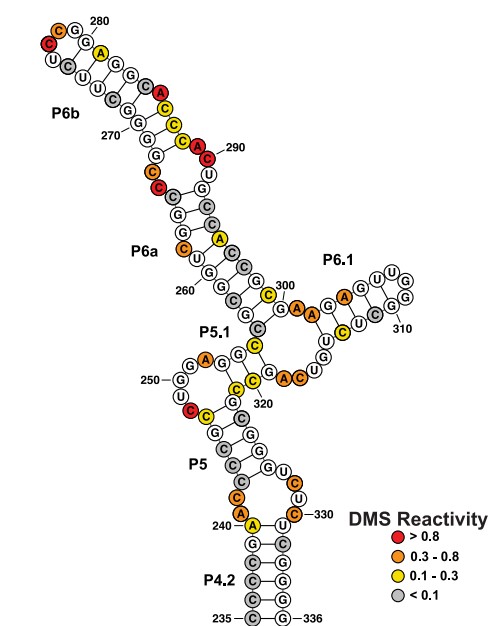

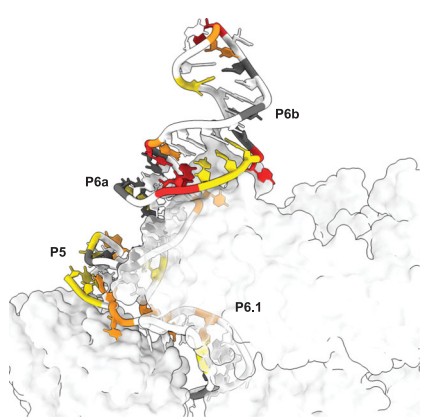

**Fig. 3 | Population average DMS reactivity of the hTR CR4/5 domain.**
**a** Normalized DMS reactivity of hTR CR4/5 domain (nt. 235–336, $n = 4$ biological replicates). Intensity of DMS reactivity colored according to the provided legend. Data are presented as means ± SD, open circles representing individual replicate values. Blue arcs designate the base pairing pattern of the canonical t/PK conformation seen by cryo-EM of assembled telomerase. **b** Secondary structure of the canonical CR4/5 conformation with DMS reactivity of the hTR population average overlaid onto the nucleotides. **c** DMS reactivity overlaid onto the cryo-EM model of assembled telomerase. PDB 7BG9. Source data are provided as a Source Data file.

moderate reactive of nucleotides within the P5.1, P6a, P6b, and P6.1 stems (Fig. 3b, c). Previously, we reported that uncharacteristic chemical reactivity of CR4/5 in vitro originates from its heterogeneous folding behavior[45]. Thus, we sought to determine whether the observation of base-paired nucleotides with unexpectedly high DMS reactivities could be due to alternative folding conformations of the t/PK or CR4/5 domains in the cellular environment.

**Ensemble deconvolution reveals alternative conformations of t/PK and CR4/5 domains**

Because MaPseq exploits the read-through behavior of multiple DMS modifications per RNA, it permits the use of bioinformatic deconvolution algorithms to identify distinct clusters of DMS reactivities that distinguish alternative RNA conformations[41,46,47]. We applied the Deconvolution of RNA Ensembles by Expectation Maximization (DREEM) algorithm[41] to our DMS reactivities of endogenous hTR. We then used DMS reactivities derived from the clusters to guide thermodynamic secondary structure predictions. We found that the hTR t/PK domain forms distinct major and minor clusters across multiple biological replicates ($n = 4$) with relative abundances of 79.3 ± 4.9 % and 20.8 ± 4.9 %, respectively (Supplementary Fig. 1). Nucleotides responsible for forming the pseudoknot P3 helix (C108, A111, C112 and A174-A176, C180, A181, C183) are overall weakly DMS reactive in the major cluster (Fig. 5a), while in the minor cluster they partition into strongly reactive (C108, A111, C112, C180, A181, C183) and mostly unreactive (A174-176) groups.

Cluster-guided thermodynamic structure prediction of the major cluster typically yielded a conformation resembling the canonical t/PK domain, but with a slightly truncated pseudoknot P3 helix. In contrast, the model predicted for the minor cluster lacked the pseudoknot entirely and showed a substantially rearranged, extended stem-loop architecture (Fig. 5b, c). Notably, in one replicate experiment, neither cluster predicted the canonical-like t/PK conformation, underscoring the challenges associated with consistently predicting RNA pseudoknot folds using the RNAstructure software, even when guided by experimental data[48]. Across four independent replicates, the AUROC values calculated for the major DREEM cluster DMS reactivity patterns against the data-driven, canonical-like t/PK domain structure prediction averaged 0.86 ± 0.01 (Supplementary Fig. 2). Similarly, the AUROC analysis for the minor DREEM cluster reactivity patterns against the alternative t/PK domain structure prediction yielded an average of 0.88 ± 0.08. The slightly lower AUROC value for the major cluster's canonical-like t/PK structure model likely stems from the RNAstructure folding algorithm's limitations in accurately predicting pseudoknot structures. Moreover, the variability observed in the AUROC values for the minor cluster's alternative t/PK structure model suggests that DREEM is not able to fully deconvolute the heterogeneity across the t/PK domain. This conclusion is reinforced by the observation that DMS reactivity patterns for minor DREEM clusters did not correlate as strongly between replicate experiments as those for the major DREEM clusters (Supplementary Fig. 1b). The lower correlation between cluster replicates may be attributed to subtle additional structural heterogeneity within the t/PK domain, which poses challenges for the

Despite the good agreement of our DMS reactivity data with current hTR structural models, we noted the presence of numerous nucleotides that are predicted to be base paired but show low (0.1–0.3) to moderate (0.3–0.8) DMS reactivities. For example, within the t/PK domain the P2, P2a, and P2a.1 stems, as well as the pseudoknot forming P3 stem, all contained nucleotides with varying degrees of reactivity (Fig. 2b). Similarly, analysis of the CR4/5 domain revealed low to

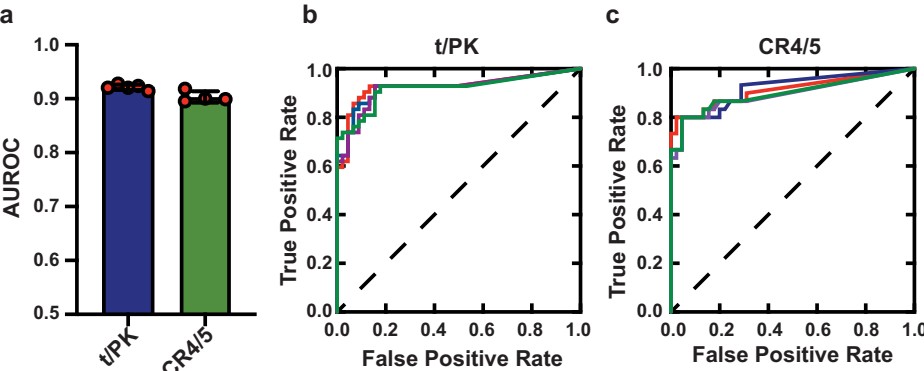

**Fig. 4 | Evaluating data-driven RNA structure models with the area under the receiver operating characteristic curve (AUROC). a** Bar plot depicting AUROC values for DMS reactivities derived from five biological replicates of the t/PK domain (left, blue bar) and four biological replicates of the CR4/5 domain (right, green bar). Data are presented as means ± SD. **b** Receiver operating characteristic (ROC) curve comparing the t/PK Chen model with our DMS reactivities of the t/PK domain from four replicates (different colored lines). **c** ROC curve comparing the CR4/5 Chen model without DMS reactivities of the CR4/5 domain from four biological replicates (different colored lines). Source data are provided as a Source Data file.

DREEM approach to fully resolve, possibly due to limited information from co-occurring mutations in adenine and cytosine bases[49].

The CR4/5 domain also formed reproducible major and minor reactivity clusters across multiple independent replicates (n = 4), with relative abundances of 87.0 ± 1.4 % and 13.0 ± 1.4 %, respectively (Supplementary Fig. 3a). However, in contrast to our results with the t/PK domain, the CR4/5 reactivity clusters produced by DREEM were highly correlated across experimental replicates for both the major and minor populations (Supplementary Fig. 3b). Two cytosines that form the internal loop bisecting P6a and P6b (C266-267) are strongly reactive in the major cluster but are moderately reactive in the minor cluster (Fig. 6a). An additional three cytosines that form P6b (C286-288) are weakly reactive in the major cluster but strongly reactive in the minor cluster. Structure prediction of the major cluster yields the canonical three-way junction CR4/5 architecture (Fig. 6b). The minor cluster is predicted to form a linear, register-shifted stem loop, with strongly reactive P6a nucleotides (A285-C290) forming the apical loop (Fig. 6c). Notably, this alternative conformation of CR4/5 lacks the P6.1 stem loop, which is essential for telomerase assembly and catalytic activity. This predicted alternative CR4/5 conformation is unlikely to arise from an artifact of DREEM clustering, as it is a predicted structure with similar free energy to the canonical three-way junction form even in constraint-free structure prediction runs (Supplementary Fig. 4). Across four independent replicates, the AUROC values calculated for the major DREEM cluster DMS reactivity patterns evaluated against the data-driven canonical CR4/5 domain major structure prediction yielded a mean value of 0.97 ± 0.002. The AUROC analysis for the minor DREEM cluster reactivity patterns evaluated against the alternate register-shifted CR4/5 domain structure prediction yielded a mean value of 0.88 ± 0.009 (Supplementary Fig. 2). Taken together, the AUROC analysis for the CR4/5 domain shows a high degree of reproducibility for the canonical and alternative data-driven RNA structure models.

We note that while major and minor clusters are observed for both the t/PK and CR4/5 domains, our sequencing library preparation method splits the two domains prior to DREEM deconvolution, and therefore does not provide information regarding whether the two minor conformations reside in the same molecule. Moreover, AUROC analysis of the modeling results for the t/PK domain indicates a higher level of structural heterogeneity for this domain and suggests the presence of more conformations in the structural ensemble than can be resolved by our deconvolution analysis. In contrast, the high-quality modeling and the excellent reproducibility of the reactivity patterns

for the alternative conformation of the CR4/5 domain prompted us to focus on the functional impacts of folding heterogeneity within this domain.

## Overexpression of hTERT does not cause refolding of the CR4/5 domain of hTR

The observation that the majority of the hTR CR4/5 domain is folded into the canonical structure in vivo raised the question of whether this cellular fraction corresponds to RNA that is co-assembled with the hTERT catalytic protein subunit. To test this hypothesis, we performed DMS-MaPseq experiments using BJ fibroblasts, a cell line that constitutively expresses hTR but does not express the hTERT protein subunit[50]. Surprisingly, DMS-MaPseq and DREEM deconvolution of endogenous hTR in BJ fibroblasts revealed proportions of the major canonical (86%) and the minor alternative (14%) conformations of the CR4/5 domain similar to hTR in HeLa cells across two biological replicates (Supplementary Figs. 5, 6). Thus, hTR folds into the two CR4/5 structures independent of hTERT.

As a separate test of the dependence of our DREEM deconvolution results on hTERT expression levels, we next performed DMS-MaPseq and DREEM deconvolution on HeLa cells that were transiently transfected with a plasmid that overexpressed hTERT protein. Once again, the DREEM deconvolution results remained highly similar, revealing major canonical (85%) and minor alternative (15%) conformations of the CR4/5 domain (Supplementary Figs. 6, 7). However, we noticed a lower degree of correlation at the level of single nucleotide DMS reactivity values between clusters derived from HeLa cells overexpressing TERT and BJ fibroblasts. (Supplementary Figs. 6b, c). This low degree of correlation is likely an effect of the presence of hTERT, as evidenced by comparatively higher correlation of DREEM clusters between TERT positive cell lines (HeLa and HeLa + hTERT) versus TERT negative cell lines (BJ fibroblast) (Supplementary Fig. 8). Even so, the relative abundance of the canonical and alternative CR4/5 conformations between all three cellular contexts are near identical (Supplementary Fig. 6a). Taken together, these results demonstrate that the major hTR CR4/5 cluster that exhibits the canonical structural fold is not dependent on the presence and binding of hTERT protein.

## CR4/5 mutations can specifically stabilize the alternative conformation

To further explore the biological effect of the linear alternative CR4/5 on telomerase biogenesis, we aimed to bias the hTR folding landscape toward this conformation in vivo and query its effects on telomerase

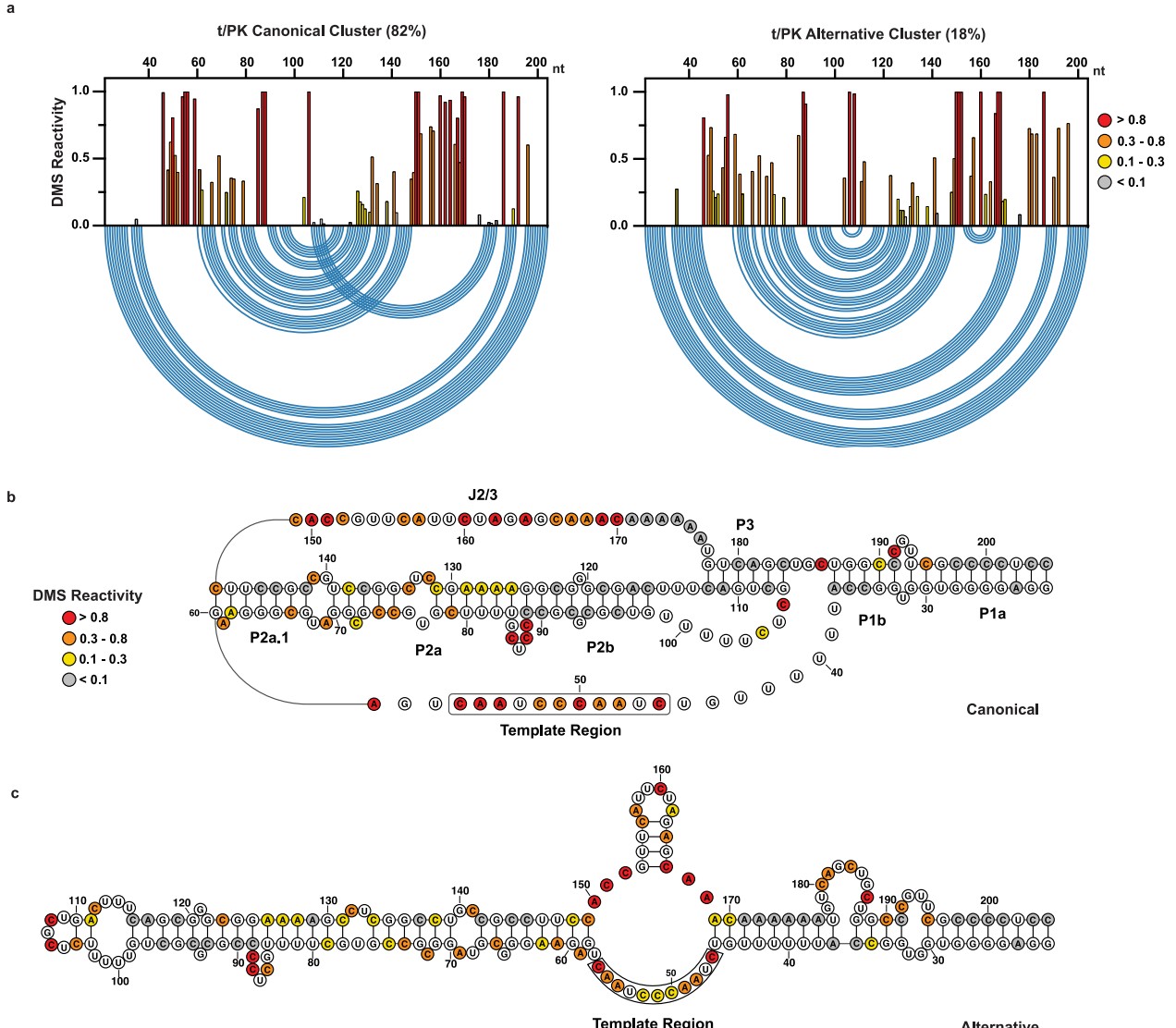

**Fig. 5 | DREEM-deconvoluted DMS profiles and structure predictions of hTR t/PK. a** Normalized DMS reactivity profiles of the two clusters predicted by DREEM. Intensity of DMS reactivity colored according to the provided legend. Blue arcs designate the base pairing pattern of the data-guided predicted t/PK secondary structure. **b** Data-guided secondary structure prediction of the t/PK domain from the canonical cluster of DMS reactivities. **c** Data-guided secondary structure prediction of the t/PK domain from the alternative cluster of DMS reactivities. Source data are provided as a Source Data file.

RNP assembly and catalytic activity. We identified three different nucleotide positions within the CR4/5 P6b stem that, when mutated, are predicted to disrupt base pairing interactions in the canonical conformation and/or introduce stabilizing base pairs in the alternative conformation without directly disrupting interactions with hTERT (Fig. 7a, b). Our designed hTR mutant constructs contain either one (M1: G270C), two (M2: C277G, A281U), or three (M3: G270C, C277G, A281U) nucleotide substitutions in the CR4/5 domain.

Each of the hTR variants was sub-cloned into a plasmid under the control of the U1 snRNA promoter and transfected into HeLa cells together with a separate plasmid expressing hTERT with an N-terminal FLAG tag. Following each transfection, we performed the live cell DMS MaPseq procedure to analyze the folding properties of each transiently overexpressed hTR variant. Across two independent biological replicates, the resultant DMS profiles of all mutants displayed elevated DMS reactivity at nucleotides C286-288, consistent with an increased relative abundance of the alternative CR4/5 conformation compared to the canonical conformation (Fig. 7c). Corroborating this finding, DREEM deconvolution followed by structure prediction revealed an increased representation of the alternative CR4/5 conformation in the ensemble. The CR4/5 structural ensemble of M1 and M2 is ~45% folded in the alternative conformation, while M3 adopts the alternative conformation with ~90% abundance, representing a drastic redistribution of the CR4/5 structural ensemble away from the canonical conformation (Fig. 7c). We note that the redistribution of molecules between the two conformations in response to these mutations provides additional evidence supporting the structural models. As a control, we coexpressed wild type hTR with N-FLAG hTERT and analyzed its structure. Interestingly, overexpressed wild type hTR had an increased proportion of alternatively folded CR4/5 compared to endogenous wild type hTR (27 ± 8% and 13 ± 1.4% %, respectively) (Figs. 7c and Supplementary Fig. 9). This is potentially an effect of increased hTR concentration in the nucleus overwhelming some component (such as a protein chaperone) that aids hTR folding.

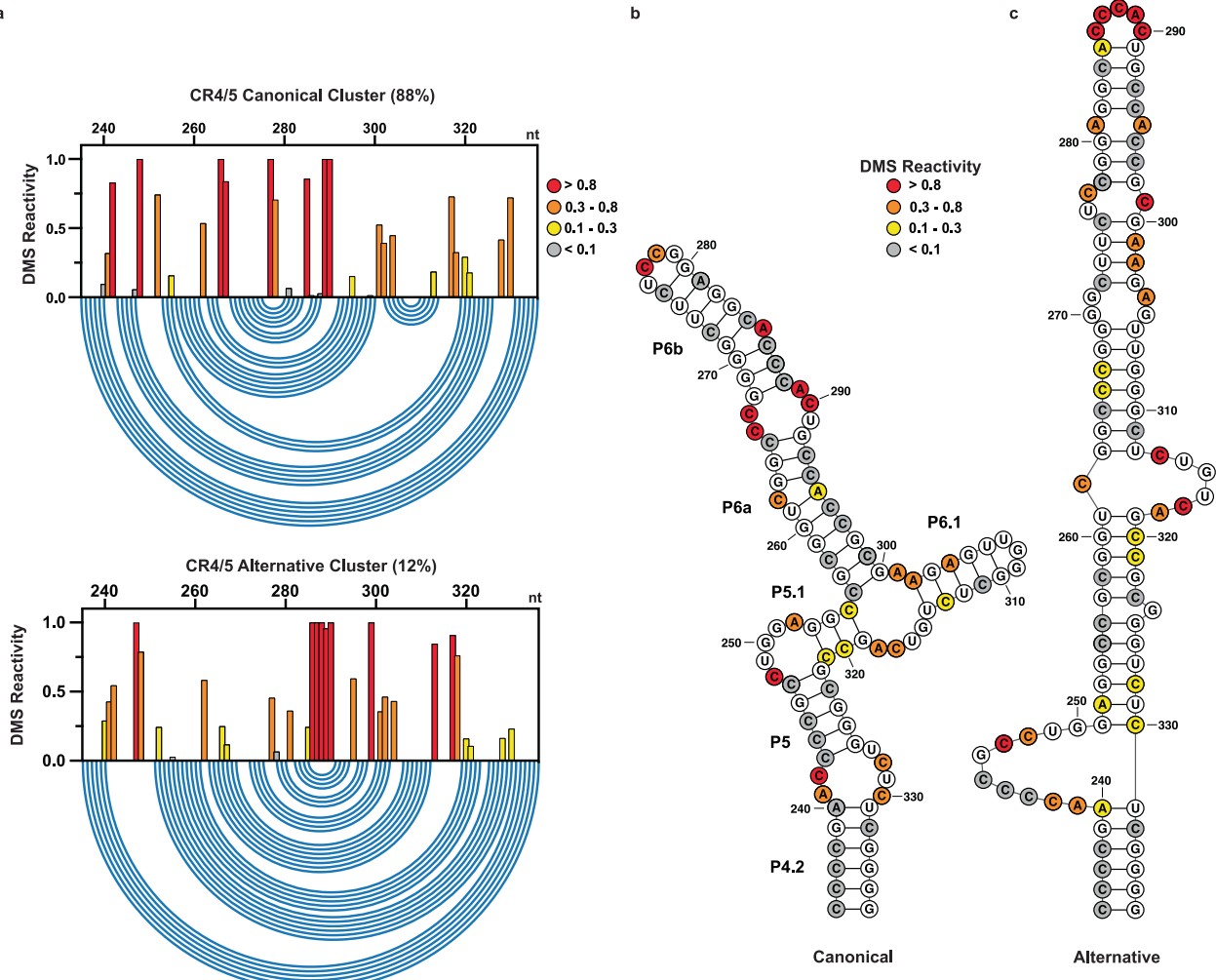

**Fig. 6 | DREEM-deconvoluted DMS profiles and structure predictions of hTR CR4/5. a** Normalized DMS reactivity profiles of the two clusters predicted by DREEM. Intensity of DMS reactivity colored according to the provided legend. Blue arcs designate the base pairing pattern of the data-guided predicted CR4/ 5 secondary structure. **b** Data-guided secondary structure prediction of the CR4/5 domain from the canonical cluster of DMS reactivities. **c** Data-guided secondary structure prediction of the CR4/5 domain from the alternative cluster of DMS reactivities. Source data are provided as a Source Data file.

## The alternative CR4/5 conformation does not support efficient telomerase assembly and activity

Having measured the increased abundance of the alternative CR4/5 conformation in cells transfected with the mutant hTR variants, we next set out to investigate the impact of the observed change in CR4/5 folding on telomerase RNP assembly and catalytic activity. For these experiments we transiently transfected either HeLa or HEK293T cells with the WT and mutant hTR constructs together with FLAG-tagged hTERT. Following immunopurification of the telomerase RNPs from each transfection experiment via the FLAG-tagged hTERT subunit, we performed direct primer extension activity assays on the IP fraction. M1 and M2 each exhibited moderately reduced total activity compared to WT (~70–80% activity) while M3 was severely deficient (~5% activity) (Fig. 8a–c). We note that the residual catalytic activity observed for the M3 experiment may result from assembly of endogenous WT hTR with FLAG-tagged hTERT.

Importantly, western blots for FLAG-hTERT showed that perturbation of the CR4/5 folding landscape did not alter the steady-state amount of hTERT (Fig. 8d, **top panel**). Thus, the reduction in telomerase catalytic activity cannot be explained by reduced stability of the hTERT subunit. Furthermore, northern blots for hTR showed similar levels in WT and mutant cell extracts when normalized to both

U1 and U2 snRNAs (Fig. 8e, **top panel**), so the reduction in telomerase activity cannot be explained by reduced levels of mutant hTR. Interestingly, hTR migrated as single band when samples were not heated prior to loading on the gel, whereas a previously reported hTR doublet[40] was observed when samples were preheated and snap-cooled. This result suggests that heating and quickly cooling the RNA sample prior to loading on the gel traps hTR conformations that are not completely resolved on the denaturing gel.

We next asked if the reduced telomerase activity reflected less active assembled RNP enzymes or, rather, a failure to assemble RNP complexes. When telomerase was immunopurified by the FLAG epitope tag on the hTERT protein, northern blotting for hTR in the IP fractions revealed a substantial decrease of M3 hTR relative to WT, M1 and M2 hTR (Fig. 8e, **bottom panel**). As a further test of whether each hTR variant was competent to assemble into a functional telomerase RNP complex, we performed western blots for the dyskerin (DKC1) and Gar1 subunits of the H/ACA RNP complex (Fig. 8c, bottom two panels). These proteins serve as surrogate markers for the hTR RNA, because they should only occur in the anti-hTERT IP by virtue of their binding to hTR. Our results showed that overexpression of the M1 and M2 RNAs resulted in modest reduction in both DKC1 and Gar1 levels in the IP fraction, while M3 reduced DKC1 and Gar1 levels in the IP much more

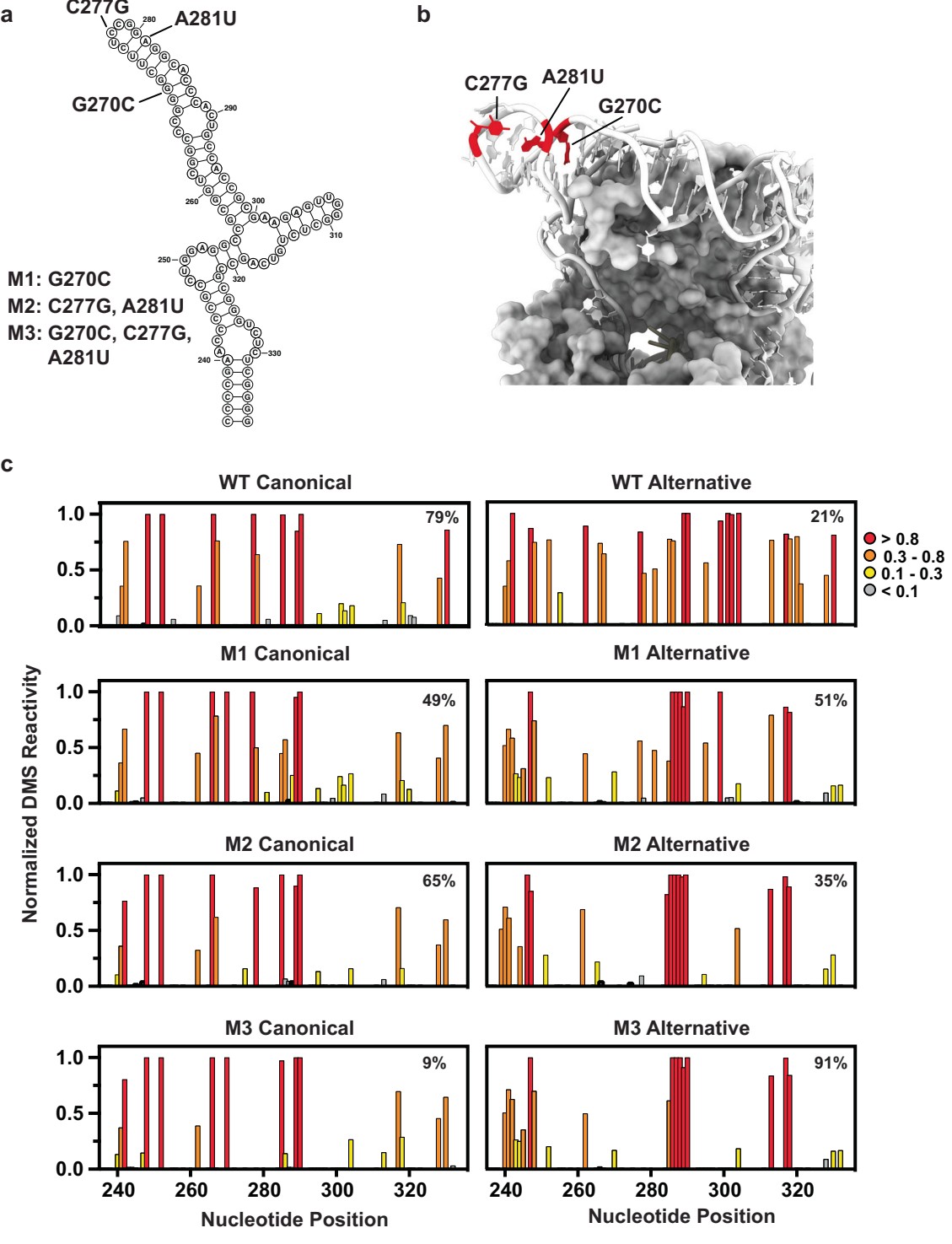

**Fig. 7 | Design and MaPseq validation of CR4/5 mutants. a** Canonical CR4/5 secondary structure with mutated nucleotides labeled. **b** Cryo-EM structure (PDB 7BG9) of CR4/5 with mutated nucleotides highlighted. **c** Normalized DMS reactivity profiles of clusters predicted by DREEM for overexpressed WT and CR4/5 mutants. Source data are provided as a Source Data file.

drastically. Since DKC1 and Gar1 bind to the scaRNA domain of hTR, they would be expected to copurify only with FLAG-hTERT that was bound to hTR; thus, these results provide further evidence for the inability of the M3 hTR variant to co-assemble with TERT. Taken together, these results support the conclusion that the M3 hTR variant, which predominantly forms the alternative CR4/5 fold in vivo, has substantially decreased telomerase RNP assembly efficiency.

## Telomerase RNP purified from cells lacks the alternative CR4/5 conformation

Because forcing the alternative CR4/5 conformer inhibited telomerase assembly and activity, we hypothesized that the endogenously assembled telomerase would not contain the alternative hTR conformer. To test this, we immunopurified telomerase from HEK293T cells that were transiently transfected with plasmids

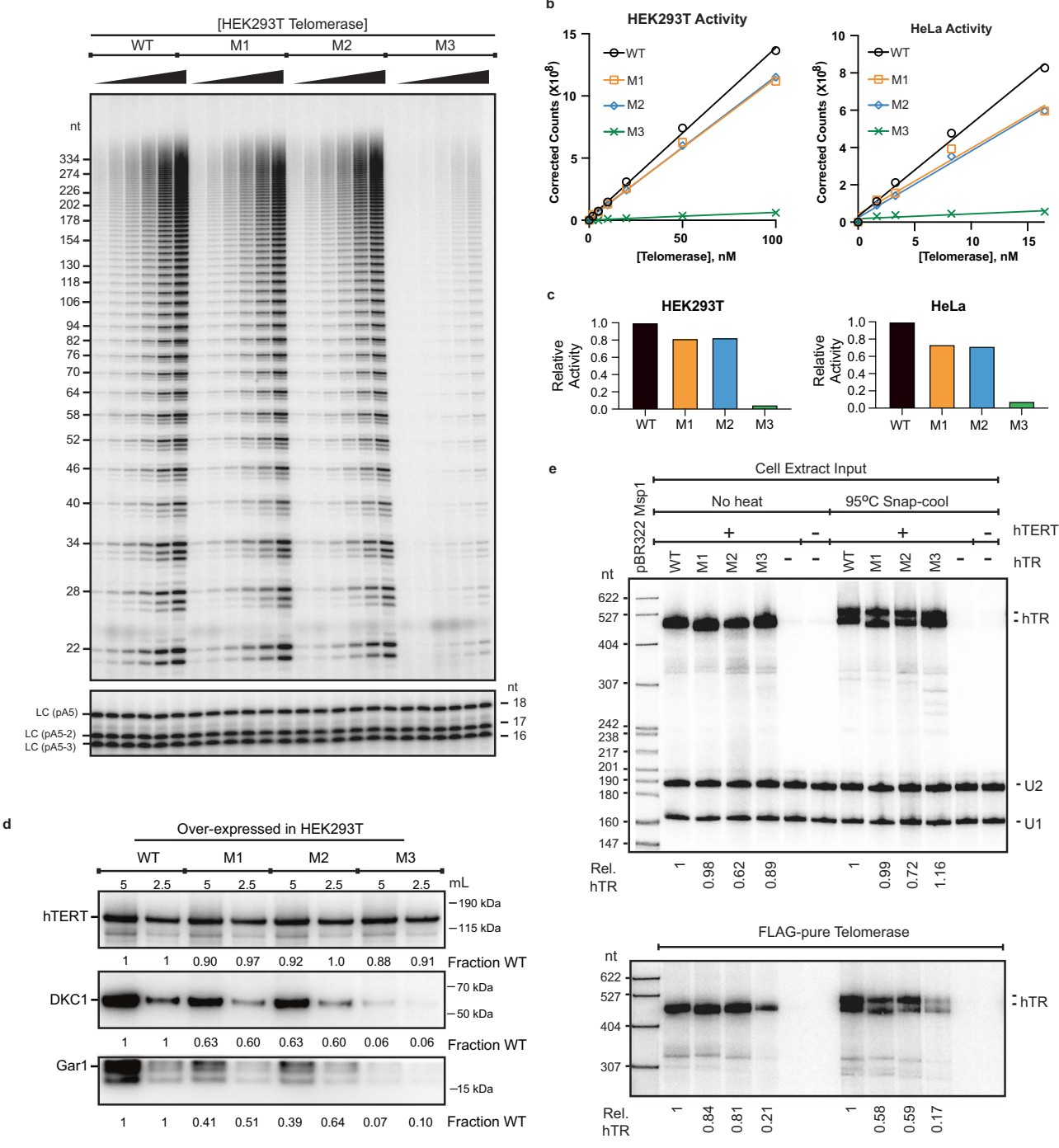

**Fig. 8 | Telomerase activity and RNP assembly for hTR mutants. a** Activity assay of FLAG-purified HEK293T-derived telomerase measuring incorporation of $^{32}$P-dGTP into a telomeric DNA primer for the WT, M1, M2, and M3 hTR variant constructs. LC, labeled oligonucleotide loading controls. Experiment was repeated four times ($n = 4$) with similar results. **b** Quantification of telomerase activity. HEK293T- or HeLa-derived telomerase activity calculated by summing the total lane signal and correcting for differences in intensity of the loading controls. **c** Bar plot of fractional activity relative to WT was calculated from the slopes of the linear regression fits. **d** Western blots for hTERT (top), DKC1 (middle), and Gar1 (bottom)

all performed after IP of FLAG-hTERT. Relative signal compared to wild type for two different loading amounts is shown below each lane. (Gar1 values are approximate due to accidental truncation of tops of bands.) Experiment was conducted two times ($n = 2$) with similar results. **e** Top, northern blot for hTR in HEK293T cell extracts, each hTR intensity normalized to U1 and U2 snRNA recovery controls and then to the WT hTR signal. Bottom, northern blot for hTR in anti-hTERT IP fractions, values normalized to WT. hTR migrates as a doublet when samples are heated and snap-cooled (right lanes) because the 7 M urea gel is not completely denaturing. Experiment was conducted four times ($n = 4$) with similar results.

expressing full-length hTR and FLAG-tagged hTERT[51]. Visual inspection of the measured DMS reactivities of the CR4/5 probed within purified telomerase RNP complexes with recent cryo-EM derived models of assembled telomerase[19] revealed several regions of reactivity that appear to be within the binding interface of hTR with the hTERT RNA

binding domain (RBD) (Supplementary Fig. 10). For example, the helical bulge regions that includes C266, C267, A289, and A290 all show medium to high reactivity (Supplementary Fig. 10a, b) but are in close proximity to the protein binding interface in the structure (Supplementary Fig. 10).

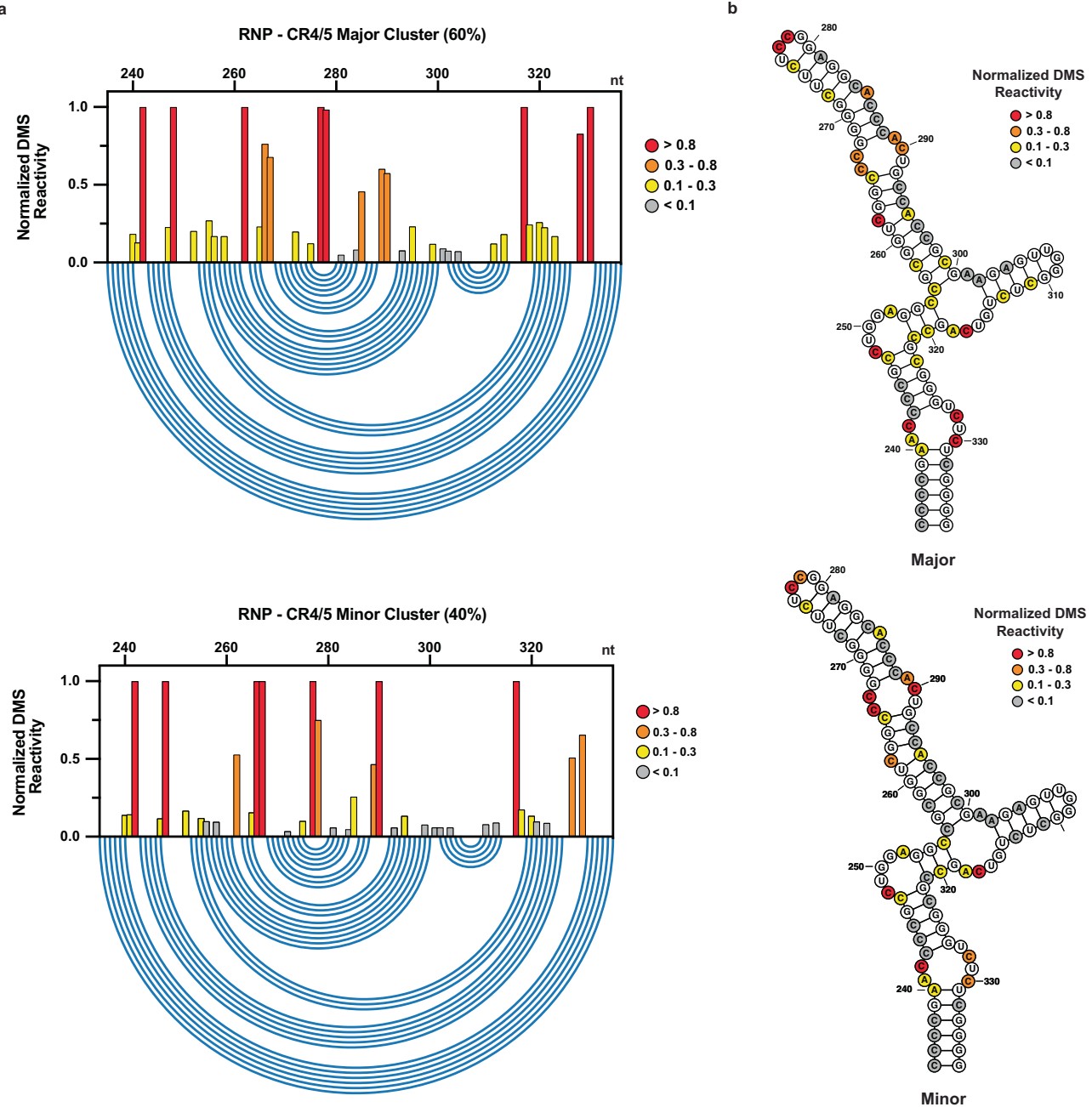

**Fig. 9 | DREEM-deconvoluted DMS profiles and structure predictions of hTR CR4/5 from telomerase RNP complexes assembled in cells and purified via the FLAG-tag on hTERT. a** Normalized DMS reactivity profiles of the two clusters predicted by DREEM. Intensity of DMS reactivity colored according to the provided legend. Blue arcs designate the base pairing pattern of the data-guided predicted CR4/5 secondary structure. **b** Data-guided secondary structure prediction of the two clusters with overlaid DMS reactivities. The major and minor clusters have identical structures. Intensity of DMS reactivity colored according to the provided legend.

We next performed DREEM deconvolution on the DMS-MaPseq data collected on purified telomerase RNP complexes and found that two different reactivity clusters emerged (Fig. 9a). Analysis of experiments performed at either 1% or 2% DMS (v/v) revealed consistent DREEM deconvolution results, producing major and minor reactivity clusters of $62 \pm 2\%$ and $38 \pm 2\%$, respectively. Interestingly, the two cluster reactivity-driven structure predictions were identical and matched the canonical CR4/5 fold, whereas the previously characterized linear alternative CR4/5 fold was not observed (Fig. 9b, c). We noted that the aforementioned P6 stem helical bulge nucleotides (C266, C267, A289, and A290) are less reactive in the

major cluster (60%) than in the minor cluster (40%). Thus, we suggest that DREEM deconvolution is reporting on RNP structural heterogeneity, wherein complexes with a tight protein-RNA interaction in the hTR P6 helical bulge region are more protected from DMS modification. The observation of multiple clusters of DMS reactivity from HEK293T cell assembled and purified telomerase complexes is consistent with cryoEM studies that revealed substantial sample heterogeneity during particle classification[23,24]. In any case, it was reassuring that DMS-MaPseq of purified, catalytically active telomerase RNP predicted a single secondary structure that agreed with the cryo-EM structure.

## Discussion

Certain non-coding RNAs, like riboswitches, have multiple secondary structures required for their function[52]. In contrast, hTR structure is known to adopt a phylogenetically conserved secondary structure[12] and therefore would not be predicted to adopt long-lived alternative conformations in vivo. Thus, the observation that 12–18% of cellular hTR exists as a misfolded conformer was unexpected. The propensity of RNA to fall into kinetic traps during folding is well established in vitro; however, a reasonable expectation would be that such misfolded RNAs would be subject either to refolding or to degradation in vivo[53–56]. Instead, we found that alternative conformations of the t/PK and CR4/5 – conformations that are inconsistent with the cryo-EM models of telomerase – persist in the steady state at endogenous levels of expression. If the alternative hTR conformer is being refolded or degraded, the rates of such processes must be slow enough to permit a substantial misfolded population to exist in the steady state. On the other hand, it also remains possible that the alternative hTR conformer could exist in a different RNP with a non-telomerase function, though this notion is speculative and would require future exploration.

We tested whether hTR that adopted the alternative CR4/5 structure was in fact catalytically incompetent in vivo, as expected from previous work on the function of the three-way junction. Using structure-guided RNA mutagenesis, we were able to bias the hTR folding landscape away from the canonical fold and toward the alternative conformation. The hTR variants that modestly altered the hTR folding landscape to favor the alternative conformation (M1 and M2) supported substantial levels of telomerase assembly and ~75% catalytic activity. In contrast, the M3 hTR variant resulted in the majority of cellular hTR adopting the alternative conformation, largely eliminating RNP assembly and telomerase activity.

In vitro biochemical and biophysical analyzes of the hTR CR4/5 domain show that the RNA primarily folds into a heterogeneous ensemble of states[45]. Moreover, reconstitution of hTR with hTERT in rabbit reticulocyte lysates chaperones the RNA into a fold that is consistent with the recent cryo EM structures of the endogenously assembled RNP. In HeLa cells, the RNA subunit of telomerase is in stoichiometric excess of hTERT[17]. Furthermore, BJ fibroblasts express

hTR but not hTERT. Thus, our observation that the majority of hTR in the cell, some of which is likely not in complex with hTERT, adopts a canonical CR4/5 conformation suggests that proteins other than hTERT promote functional hTR folding. This hypothesis is further supported by our observation in HeLa cells overexpressing hTERT. The discovery of a histone H2A/H2B dimer directly bound to CR4/5 in recent cryo EM models[19,22,24] provides a strong candidate for such an RNA chaperoning activity. In fact, H2A/B could potentially perform a function analogous to that of the *Tetrahymena* telomerase holoenzyme protein p65, which induces an RNA conformational change to promote RNP assembly[57–61]. Modest stabilization of the alternative CR4/5 fold potentially allows the M1 and M2 mutants to be refolded by nuclear chaperones, explaining why they have only small effects on telomerase assembly and catalytic activity. In contrast, the M3 hTR variant, which almost exclusively adopts the alternative CR4/5 conformation, presumably has a higher energetic barrier for RNA refolding and thus is unable to be rescued.

Our characterization of the alternative pseudoknot-lacking conformer of the t/PK domain builds upon previous studies of its heterogeneous folding in human[39,62] and ciliate telomerase RNA[63–66]. We observed high and moderate levels of DMS reactivity in template sequence (nt 46-57) of both of our predicted t/PK conformations. This result suggests that assembly with hTERT is primarily associated with the formation of the pseudoknot-forming P3 helix and does not significantly reorganize the template region as was previously suggested[39]. Future experiments designed to manipulate the t/PK RNA folding ensemble will yield a richer understanding of how conformational changes of this domain promote telomerase biogenesis.

More generally, we demonstrate the efficacy of using in-cell chemical probing in combination with bioinformatic deconvolution to parse individual RNA conformations of a noncoding RNA. Our findings lead us to speculate that heterogeneous hTR folding presents a barrier in the telomerase assembly pathway. This barrier may be overcome via the action of RNA folding chaperones that rescue improperly folded hTR molecules, providing a possible mechanism for regulating telomerase biogenesis pathway and catalytic activity (Fig. 10). These same or similar techniques could be used to explore the populations of other

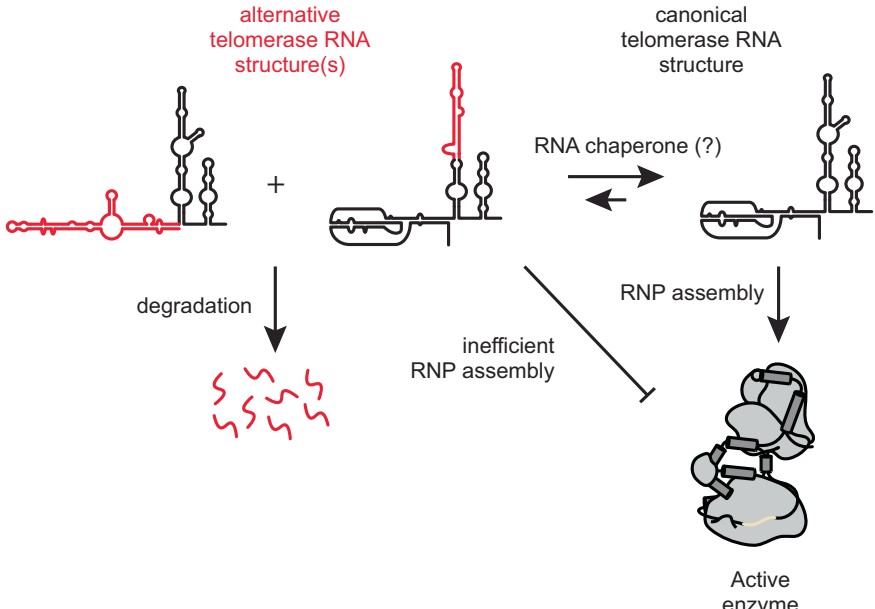

**Fig. 10 | Model of telomerase RNP assembly with respect to hTR folding.** hTR exists as a structural ensemble in cells, with a minority population adopting alternative conformations in the t/PK and CR4/5 domains (red). The canonical fold of hTR may be favored by the presence of RNA folding chaperones. hTR molecules

adopting the canonical fold are efficiently co-assembled into a functional RNP, whereas hTR molecules in the alternative conformation(s) are not efficiently assembled in the RNP complex or are slowly degraded.

functional noncoding RNAs in cells as recently done for the noncoding 7SK RNA[47]. It will be interesting to determine if the substantial sub-population of misfolded RNA seen for hTR is specific to telomerase or perhaps occurs more generally with long noncoding RNAs.

## Methods

### Cell culture and in-cell DMS modification

HeLa and HEK293T/17 cells (ATCC CRL-11268, lot:70040949) were cultured in DMEM containing 10% fetal bovine serum, 100 units/mL of Penicillin-Streptomycin, 100 mg/mL of streptomycin, and 1X Gibco GlutaMAX at 37 °C and 5% CO2. BJ fibroblast cells were cultured in the same media without GlutaMAX. Cells were seeded into 6-well plates at 0.3 ×10$^6$ cells per well and allowed to grow to 90% confluency prior to use. For DMS modification, plates of HeLa cells were divided into DMS and control groups (3 wells each). Media from the DMS group was aspirated and replaced with 2 mL of a 2% DMS (Sigma) solution in cell media (prepared by brief vortexing to incorporate DMS). Cells were incubated at 37 °C for 4 min, media was aspirated from each well, and replaced with 2 mL of ice cold DMS quench solution (30% beta-mercaptoethanol in phosphate-buffered saline). Plates were placed on ice and the cells were collected by scraping and brief trituration, then transferred to 15 mL conical tubes and resuspended in a final volume of 10 mL DMS quench solution. The cells were then pelleted by centrifugation at 200 x g for 5 min, washed once with cold PBS and total RNA harvested with 1 mL Trizol (Invitrogen) following the manufacturer's protocol.

### DMS modification of purified telomerase RNP in vitro

Aliquots of immunopurified telomerase RNP (5 μL volume) were thawed on ice, mixed with 49 μL of probing buffer (50 mM Tris-HCl pH 8.3, 50 mM KCl, 3 mM MgCl2, 10 mM DTT) and incubated at 37 °C for 20 min. DMS was then added to the samples at a final concentration of 1% or 2% and incubated at 37 °C for 5 min. To quench the DMS modification reaction, 50 μL of beta-mercaptoethanol was added to DMS and control samples. RNA from all samples was then isolated and purified by Trizol extraction, following the manufacturer's protocol.

### Transient transfection and purification of human telomerase

Plasmids containing hTERT (pvan107-3xFLAG) and WT or mutant hTR (pBSU1-hTR) were transfected at a 1:3 molar ratio using lipofectamine 2000 (11668019, Thermo Fisher Scientific). The cells were further expanded 3-fold 24 h after transfection and then 24 h later either harvested or used in DMS probing reactions. To purify telomerase, the cell pellet was lysed with CHAPS lysis buffer (10 mM Tris–HCl pH 7.5, 1 mM MgCl$_2$, 1 mM EGTA, 0.5% CHAPS, 10% glycerol, 5 mM beta-mercaptoethanol,) for 45 min at 4 °C on a rotator. The lysate was then clarified by centrifugation at 13,000 x g at 4 °C for 30 min. Anti-FLAG resin (A2220, Sigma-Aldrich) was added to the clarified supernatant and the samples incubated on a rotator for 4 h (or overnight) at 4°C. The anti-FLAG resin was washed 3x with wash buffer (20 mM HEPES–NaOH pH 8.0, 2 mM MgCl$_2$, 0.2 mM EGTA, 0.1% NP-0.1% NP-0.4, 10% glycerol, 1 mM DTT) before elution using wash buffer supplemented with 0.25 mg/ml 3xFLAG peptide (F4799, Sigma-Aldrich). Purified tel-omerase complex was verified by western blotting.

### Preparation of mutant hTR plasmids

Starting with the pBS-U1-hTR plasmid[51] three hTR mutants (M1: G270C, M2: C277G & A281U, M3: G270C & C277G & A281U) were generated using the mega primer method with gel extraction of intermediate products after the first PCR step[67].

### Preparation of hTR-targeted MaPseq libraries

10 μg of HeLa cell derived total RNA was DNAse treated in 1X TURBO DNAse buffer with 1 μL TURBO DNAse enzyme (Thermo Fisher Scientific), followed by column-purification with DNA Clean and

Concentrator-5 columns (Zymo Research) following the manufacturer protocol. 5 μg of DNAse-treated RNA was mixed with 5 pmol hTR targeting primer containing a 4-nucleotide unique molecular identifier (UMI) in a volume of 11 μL, heated to 75 °C for 3 min and annealed at 35 °C for 15 min. Next, 4 μL of 5X M-MLV buffer (Promega), 1 μL 0.1 M DTT, 1 μL RNAsin Plus (Promega), and 1 μL TGIRT III (Ingex) were added and the mixture was incubated at room temperature for 30 min. Then, 2 μL of 10 mM dNTPs were added, and the reaction was mixed and incubated at 60 °C for 2.5 h. After RT, 1 μL of 5 M NaOH was added directly to the cDNA and incubated at 95 °C, followed by 2.5 μL of 2 M HCl and purification by DNA Clean and Concentrator-5 columns (Zymo Research) using an 8:1 ratio of DNA Binding Buffer to cDNA volume. The cDNA was mixed into a 50 μL second strand synthesis reaction containing 10 μL 5X GC Phusion Buffer (Thermo Fisher Scientific), 1 μL 10 mM dNTP, 25 pmol of a second hTR targeting primer containing a 4 nucleotide UMI, and 1 μL Phusion polymerase. The second strand synthesis reaction was incubated in a thermocycler with the following program: 98 °C 2 min, 60 °C 2 min, 72 °C 10 min. Second strand product was cleaned up with Ampure XP beads (Beckman Colter) at a 0.8:1 bead to sample volume ratio and eluted in 20 μL nuclease-free H$_2$O. Illumina sequencing libraries of two partially overlapping hTR regions (Pseudoknot: nucleotides 1-286 and CR4/5: nucleotides 193-451) were generated by two successive 50 μL PCR reactions. For the first PCR, 4 μL second strand product was added to a 50 μL PCR reaction containing primers for either the pseudoknot or CR4/5 amplicon and run with the following program: 10 cycles at 66 °C annealing temperature, followed by 10 cycles at a 60 °C annealing temperature. Amplicons were bead cleaned with a 0.8:1 ratio and went into a second PCR of 10 cycles at an annealing temperature of 65 °C to complete the sequencing adapters and multiplexing barcodes. Libraries were bead cleaned with a 0.8:1 ratio and quantified by Qubit (Thermo Fisher Scientific) and Tapestation (Agilent) for quality metrics before sequencing by 150 bp paired end reads on the iSeq100 (Illumina).

### Western blots

The presence of dyskerin and GAR1 in the WT and mutant telomerase complexes immunopurified from HEK293T cells was analyzed by western blotting. The primary antibodies were anti-hTERT (600-401-252, Rockland), anti-DKC1 (NBP3-16405, Novus) and anti-GAR1 (NBP2-31742). Secondary antibody was anti-rabbit (711-035-152, Jackson ImmunoResearch, West Grove, PA). All primary antibodies were diluted 1:1000 for blotting, and the secondary antibody was diluted 1:5000. Serial dilutions of the FLAG-immunopurified telomerase were run on denaturing polyacrylamide gels, transferred to Protran 0.45 mm nitrocellulose (10600002, Amersham), and probed with the primary antibodies followed by the secondary antibody. The western membrane was developed using SuperSignal West Pico PLUS Chemi-luminescent Substrate (34578, Thermo Scientific) and visualized using a FluorChem R imaging camera (Protein Simple). Quantification was done using AlphaView software version 3.5.0.927 (Protein Simple).

### Telomerase activity assays

Activity of the immunopurified human telomerase complex over-expressed in either HEK293T/17 (ATCC CRL-11268, lot:70040949) or HeLa cells was determined by a direct assay modified from a published protocol[68]. Telomerase concentrations were determined by SDS-PAGE electrophoresis of serial dilutions of the telomerase preparation and western blotting with an anti-FLAG antibody to detect FLAG-tagged TERT, with serial dilutions of a purified FLAG-tagged protein of known concentration run on the same gel providing a standard curve. The reaction mixture (20 μL) contained 1x human telomerase assay buffer (50 mM Tris-HCl at pH 8.0, 50 mM KCl, 75 mM NaCl, 1 mM MgCl$_2$, 5 mM 2-mercaptoethanol, 1 mM spermidine), 0.05 μM telomeric DNA primer, 0.5 mM dTTP, 3.3 mM dGTP, 10 mM dATP, and 0.33 mM $^{32}$P-dATP (3000 Ci/mmol, 1 Ci = 37GBq, PerkinElmer). Following a 1 h incubation

at 30 °C, reactions were stopped with the addition of 100 μL of 3.6 M NH$_4$OAc containing 20 μg of glycogen. Ethanol (500 μL) was added for precipitation. After incubating for 1 h at -80 °C, samples were centrifuged for 15 min at 4 °C. Pellets were washed with 70% ethanol and resuspended in 10 μL of H$_2$O followed by 10 μL of 2x loading buffer (94% formamide, 0.1x TBE, 0.1% bromophenol blue, 0.1% xylene cyanol). The heat-denatured samples were loaded onto a 10% polyacrylamide/7 M urea/1x TBE gel for electrophoresis (until bromophenol blue ran to the bottom of the gel). After electrophoresis, the gel was dried and quantified by using a PhosphorImager (Cytiva).

### hTR Northern blots

Deproteinized CHAPS extract or deproteinized immunopurified telomerase was run on a 6% polyacrylamide/7 M urea/1xTBE gel. Nucleic acid was transferred to Hybond-N+ membrane at 2 amps in 0.5x TBE at 4 °C for 1.5 h. Nucleic acid was crosslinked to the membrane using a CL-1000 ultraviolet crosslinker using 254 nm lamps with an energy level of 1200 ×100 mJ/cm$^2$. The membrane was blocked using rapid-hyb buffer for 30 min at 50 °C. 10$^7$ counts per minute of a DNA probe labeled with γ-$^{32}$P-ATP and polynucleotide kinase was added and incubated overnight at 50 °C. Membrane was washed 3x with 2x SSC/0.1% SDS for 15 min at 50 °C, followed by 1x for 15 min with 0.2x SSC/0.1% SDS. Membrane was wrapped in saran wrap and exposed to a phosphor screen (Cytiva).

### Bioinformatics and DREEM analysis

FASTQ files were trimmed of sequencing adapters and filtered remove reads smaller than 120 nt long using the TrimGalore tool (github.com/FelixKrueger/TrimGalore) with the following command: trim_galore –paired –nextera read1.fastq read2.fastq –length 120. Trimmed FASTQ files were then aligned to the mature 451 nt hTR sequence (NCBI reference Sequence: NR_001566.1) with bowtie2[66] using the following parameters: --local --no-unal --no-discordant --phred33. Reads failing to align, and those that align discordantly, are discarded. SAM files were converted to the BAM file format using SAMtools with the following command: samtools view -S -b samfile > bamfile. Clustering of DMS MaPseq reads was performed with a local install of the Detection of RNA folding Ensembles using Expectation Maximization (DREEM) software[41], publicly available on the Code Ocean repository. For each pair of aligned reads, DREEM generates a bit vector where positions matching the reference sequence are assigned a value of 0, mismatched or deleted positions are assigned a value of 1, and positions with Phred scores > 20 are considered uninformative. The population average DMS reactivity of each nucleotide was calculated by taking the ratio of mismatches and deletions over to the total number coverage. Bit vectors were discarded if they contained 50% or more uninformative positions, contained mutations exceeding 10%, contained two mutations within a span of four nucleotides, or contained mutations adjacent to an uninformative position. Bit vectors were clustered with a maximum of K = 3 clusters. The number of clusters resulting in the minimum Bayesian Information Criterion (BIC) was chosen as the most likely K value[41]. Prior to RNA structure prediction, DMS reactivities were normalized by dividing by the median of the top 10% reactivity values and capped at a maximum value of 1. DREEM analysis windows for the Pseudoknot/Template and CR4/5 domains were 22-204, and 213–368, respectively. Folding predictions were performed using default parameters of the RNAStructure package[69] with the -dms parameter to use normalized DMS reactivities as constraints in prediction. Pseudoknot prediction was performed using the ShapeKnots[48] function of RNAstructure with default folding parameters and utilizing DMS constraints in prediction.

### Generating receiver operating characteristic curves and computing AUROC

The AUROC quantifies how well DMS/SHAPE reactivities support the predicted RNA structure, under the assumption that paired bases should be less reactive than unpaired bases. Based on the secondary structure, each position was labeled as paired or unpaired, and the DMS reactivities were partitioned into paired and unpaired groups based on these labels. The ROC curves and AUROC values were computed using SciPy[70]. Here, "true" and "false" positives represent, respectively paired and unpaired nucleotides with DMS reactivities less than the sliding threshold.

### Data visualization

Bar and scatter plots were generated with Graphpad Prism. RNA secondary structures were generated with VARNA[71]. Arc diagrams were generated using R-chie[72]. Images of three-dimensional molecular models were generated using UCSF ChimeraX[73].

### Statistical analysis for correlation and significance

Pearson correlation and statistical significance ($P$ values) of DMS reactivities between independent biological replicates or between cell types were computed using GraphPad Prism.

### Reporting summary

Further information on research design is available in the Nature Portfolio Reporting Summary linked to this article.

## Data availability

Sequencing data, normalized DMS reactivities, and predicted RNA secondary structures generated in this study have been deposited into NCBI's Gene Expression Omnibus with the GEO series accession number GSE245536 [https://www.ncbi.nlm.nih.gov/geo/query/acc.cgi?acc=GSM7844704]. Source data are provided with this paper.

## Code availability

The Detection of RNA Ensembles using Expectation Maximization (DREEM) software (v1) is available on Code Ocean at https://doi.org/10.24433/CO.0380995.v1.

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

## Acknowledgements

This work was supported by the National Institutes of Health R01GM095850 and R35GM153235 to M.D.S. T.R.C. is an investigator of the Howard Hughes Medical Institute. S.R. is supported by the Burroughs Wellcome Fund (#1018246.01). N.M.F was supported by NIH T32 GM133391.

## Author contributions

N.M.F. and J.Z.W. designed, conducted, and analyzed MaPseq experiments of endogenous hTR. N.M.F. and M.D.S designed, conducted, and analyzed MaPseq experiments of transfected hTR. A.G.J. contributed to sequencing library preparation. E.E performed and analyzed additional MaPseq experiments of endogenous hTR. A.J.Z. designed, conducted, and analyzed all immunopurifications, telomerase activity assays, western blots, and northern blots. M.D.S., S.R., and T.R.C contributed to overall experimental design, data analysis, and interpretation. N.M.F. and M.D.S. designed all manuscript figures. All authors participated in the writing of the manuscript.

## Competing interests

T.R.C. is a scientific advisor for Eikon Therapeutics, Storm Therapeutics, lincSwitch Therapeutics, and Somalogic, Inc. The remaining authors declare no competing interests.
