## [Peer Review file · Nature Communications]

Telomerase RNA structural heterogeneity in living human cells detected by DMS-MaPseq

Corresponding Author: Dr Michael Stone

Version 0:

Reviewer comments:

Reviewer #1

(Remarks to the Author)

In this manuscript Forino and colleagues analyze the structure of the Telomerase RNA component in human cells, by DMS-MaPseq coupled to DREEM ensemble deconvolution.

The story is potentially interesting and of high relevance. However, there is a number of points that the authors need to address to support their claims:

- "Our DMS reactivities of the t/PK domain (Fig. 2a and 2b) are in close accord with prior chemical probing studies". Where is this shown or quantified?
- "We detected DMS reactivity in the nucleotides comprising the P6.1 stem (A302, A304, C311, C313), a finding consistent with previous chemical probing studies". As per my previous comment.
- "DMS reactivity of the CR4/5 domain was consistent with a three-way junction architecture". This should also be quantified. Visual inspection of the data is not sufficient. You can use an AUROC for example.
- "The results of the deconvolution analysis were reproducible across multiple biological replicates". Not quite true. If I look at Supplementary Fig. S1, it looks like the deconvoluted profiles for the alternative cluster do not well correlate between replicates. I suspect this might be due to the low abundance of this conformer. What was the sequencing depth used to perform this analysis? Would increasing the coverage help? I would also suggest to make another replicate at least.
- The authors perform their analysis in HeLa cells. To really understand the significance of the identified alternative cluster, the authors should also perform a number of analyses here outlined:

1. transfect the hTERT construct in a cell line that does not naturally express any of the telomerase components
2. analyze cells in which high levels of hTERT are present, such as human embryonic stem cells
3. over-express the protein components of the telomerase complex

These experiments would help elucidating whether proteins play any role in the proper folding of hTERT.

- The authors only analyze mutations stabilizing the alternative structure. What about mutations stabilizing the accepted structure? Does that increase telomerase activity? I miss that as a proper experimental control.
- Is this alternative conformation truly a kinetic trap, and does it have any functional relevance, or is it just an evolutionary bystander/byproduct? For instance, if performing the same analysis in mouse cells, can the authors identify the same 2 conformations? Does covariation analysis support the alternative structure?

Reviewer #2

(Remarks to the Author)

The paper presents chemical probing of human telomerase RNA (hTR) in living cells using dimethyl sulfate mutational profiling with sequencing (DMS-MaPseq) and ensemble deconvolution analysis. Specifically, the hTR template/pseudoknot (t/PK) and three-way junction (CR4/5) domains are separately probed and analyzed. The approach used allows for the analysis of more than one conformation if present. The authors find that both the t/PK and CR4/5 adopt a major conformation that seems to conform to what is expected for the assembled TERT-TR and a minor conformation that could not assemble with TERT. For CR4/5, mutations were designed to test the effects of stabilization of the alternate minor conformation and assays showed that they were detrimental to telomerase assembly and activity. The authors propose that the misfolded

CR4/5 is either slowly refolded or degraded, as well as an assembly pathway for the telomerase TERT-TR core. The experiments are technically challenging due in part to the low abundance of telomerase even in cancer (HeLa) cells, and this is the first example of the modern probing techniques and ensemble deconvolution analysis being applied to telomerase RNA. It is a bit disappointing that only the CR4/5 is analyzed in detail and not t/PK, especially as a model for assembly is presented in the last figure.

Major:

1. It is not clear to me how/whether the authors distinguish free hTR from hTR assembled with TERT. According to the cited Xi and Cech (2014) paper, HeLa cells have ~240 telomerase/cell, ~1140 hTR, and ~500 TERT. For hTR, that would mean ~21% of the hTR is bound and 79% is free. In the analyzed data, the relative abundance of canonically folded and alternatively folded is 80% and 20% for t/PK and 85% and 15% for CR4/5, respectively. So if the numbers in the Xi and Cech paper are correct, these percentages cannot be for TERT bound vs free, correct? Also, wouldn't hTR bound to TERT have different reactivity from free hTR even if the fold is 'canonical'? In that case, wouldn't one expect 3 populations, free hTR 'canonical', free hTR alternately folded, and bound TR?
2. To help answer the question above, have the authors tried DMS/SHAPE+deconvolution on in vitro purified telomerase (using the FLAG-tagged hTERT purified telomerase RNA from the FLAG-hTERT and WT hTR transfected cells)? I think this control is essential, for both CR4/5 and t/PK. Also map these results on the highest resolution human telomerase structures and compare to results for the in vivo probed telomerase.
3. In the introduction, the authors mention "Overexpression of telomerase has also historically been used to mitigate problems arising from its scarcity but has been shown to bypass the endogenous RNP assembly pathway". For the CR4/5 variants, the authors overexpressed hTR and hTERT and transiently transfected to analyze their effects on structure in vivo. They did do the control probing for WT hTR (Fig S5), and say in the main text they found an increase in alternatively folded (~30% vs ~13%). However the two DMS profiles they show in the SI have ratios of 79% vs 21% and 67% vs 33%, which does not average to 30% for the 'misfolded' conformation. More importantly, the profiles look different from the results on endogenous hTR and to some extent to each other (compare Fig 5 and Fig S5). Please explain.
4. It would be useful to also see the t/PK results for overexpression vs endogenous vs in vitro purified telomerase.
5. Figure 8 "Model of telomerase RNP assembly with respect to hTR folding" and discussion completely ignores assembly with t/PK and only shows t/PK in 'canonical' conformation. Does CR4/5 bind first and then t/PK or vice versa? If one is showing assembly, why aren't the H/ACA proteins shown binding first? I suggest deleting or changing this figure.
6. In all the figures that show reactivity, the authors need to clearly distinguish between no detection and no/low reactivity. Only Cs and As seem to show any reactivity in these experiments (although DMS can be used to probe U and G).
7. Discussion "hTR structure is known to adopt a phylogenetically conserved secondary structure (ref 10) and therefore would not be predicted to adopt long-lived alternative conformations in vivo." I think I understand what the authors mean to say here, but it is not clear why an alternative structure could not form stably in absence of TERT (as proposed in the references cited in discussion, second page second full paragraph). Isn't there some evidence that hTR co-folds with TERT?
8. The authors do a nice job of referencing previous literature in the introduction, with a couple of notable exceptions. Specifically, the high-resolution human telomerase cryo-EM structures listed do not include Wan, et al Cell Res (2021) 10.1038/s41422-021-00586-7.
9. Discussion, last paragraph: "These same techniques could be used to explore the populations of other functional noncoding RNAs in cells." Haven't similar techniques been used already for lncRNAs, e.g. for 7SK RNA? 10.1016/j.molcel.2022.02.009
10. For Figure 1, the last line reads "GIVE EITHER PDB NUMBERS OR CITE REF FOR c AND d." Please correct.
11. METHODS: The methods needs to be carefully proofread. In "Cell culture and DMS modification" 100 units/mL of penicillin was used but units for streptomycin are 100 mg/m (?). Should 10 mL of ice cold DMS quench solution be 1 ml? For the Western blots and Telomerase activity assays, in several places there is m instead of (micro), incubation is at 80 °C (!). In hTR northern blots the degree sign before C is missing and "counts" should be counts per minute?
12. Please put the % of each population on Figure 4 and 5. Please provide uncertainties for the multiple replicates of the chemical probing.
13. Fig. 6 G270 should be G270C (its wrong 3 times). Also, please include results of probing on WT in this figure for comparison.

Reviewer #3

(Remarks to the Author)

In this manuscript, Forino et al., used dimethyl sulfate mutational profiling combined with sequencing (DMS-MapSeq) to probe the structure of endogenous hTR in HeLa cells. They used primers specific for the t/PK and CR4/5 domains to probe DMS reactivities for these domains of hTR. While the reactivities of most nucleotides agree with the predicted secondary

structures (SS) of hTR and cryoEM structures of human telomerase, there are a few nucleotides whose reactivities do not fit the predicted SS. The authors then used the DREEM algorithm to deconvolute their data into major and minor clusters and then used the clusters to guide thermodynamic predictions of different conformations of the tPK and CR4/5 domains. The major and minor clusters were found to correspond to the canonical conformation and alternative conformations for each of tPK and CR4/5, respectively. They chose to perform further mutagenesis experiments on the CR4/5, which showed better results from their experiments and modelling. They identified three positions in CR4/5 which were predicted to disrupt base-pairing of the canonical conformation of the RNA domain and/or stabilise the alternative conformation. They overexpressed hTR single, double or triple mutants at these positions with TERT in HeLa and performed DMS-MapSeq and DREEM deconvolution. They found that the single and double mutants showed a small increase in the abundance of the alternative conformation of CR4/5 compared to cells overexpressed with WT hTR. The triple mutant showed a large increase in the alternative conformation. They also performed immunoprecipitation on TERT and showed that the single and double mutants only showed a small reduction in the total telomerase activity and the triple mutant retained very little telomerase activity. Similarly, the amounts of dyskerin and GAR1 in the mutants were reduced moderately in the single and double mutants and drastically for the triple mutant.

Some previous studies have described alternative conformations of tPK and CR4/5 but in an in vitro context. The main novelty of the manuscript is that the authors investigated the alternative conformations of the tPK and CR4/5 domains of hTR endogenously. The effect of the triple mutant identified here is very intriguing, which would form the basis for further investigation. However, with the presented data, it is still not quite clear how exactly the alternative conformations directly contribute to telomerase assembly. Furthermore, due to the lack of detailed explanations of the methods employed, I find the manuscript a bit difficult to read and understand how the results are generated and what the pitfalls are. There are a number of specific points that the authors should consider:

1. I appreciate that the authors have included a short description of DMS-MapSeq in the introduction. It would be beneficial to provide more detailed explanations of how the methods (e.g. DMS-MapSeq, DREEM, AUROC) work and how they were specifically implemented in this study. This additional detail will enable non-specialist readers to better grasp the main points the authors aim to convey. More specific suggestions will be in some of the points below.
2. In Figures 4 and 5, the authors compare the canonical and alternative structures of hTR regions. In Figure 6, they further characterise these regions by concentrating on three residues that stabilise the alternate conformation of hTR. I noticed a few other distinctions between the canonical and alternate forms of hTR, such as tPK (194, 197-180, region around 150-170) or CR4/5 (240-242, 313-318). Is there any specific reason why these differences are not discussed or pursued with a more detailed analysis?
3. In Figures 4 and 5, there are many residues which are involved in base-pairing but exhibit considerable DMS-reactivities (4b: 62, 66, 74, 75, 79, 126-129, 138, 148, 190, 196; 4c: 35, 61, 62, 66, 74, 75, 79, 104, 123, 126-128, 131, 138, 148, 149, 164, 169, 170, 190, 196; 5b: 255, 295, 302, 304, 313, 320, 321; 5c: 240, 252, 266, 267, 285, 301, 302, 320, 321, 328, 330). How are these reactivities accounted for?
4. How do DMS reactivities of nucleotides change in response to binding to protein partners (such as TERT or histones)? I am not familiar with the field so it would be good for the authors to take this into account in their results/discussions.
5. The DREEM deconvolution yields only two conformations of each of the tPK and CR4/5 domains. I would expect that endogenous hTR would have more conformations than just two. How likely is it that more conformations may exist that could not be captured by the deconvolution method?
6. The normalisation procedure for DMS reactivity in the reads mentioned in Figures 2, 3, 4, 5, and 6 is not explicitly outlined in the methods or the text. It would be helpful for a more comprehensive understanding of the study to include details about how DMS reactivity was normalised in these figures.
7. They also calculated the area under the receiver operating characteristic curve (AUROC) to evaluate the quality of their models, which are shown in Fig. S4. I do not understand what this Figure shows and how the overall values are derived from this figure. What do the two colors in the yellow bars represent? How are the DMS modification levels shown in the blue bar normalised? It is also intriguing that the tPK canonical form has the lowest confidence level. Is there an explanation for this?
8. In the abstract, the authors argue that stabilising the alternate forms of CR4/5 has a detrimental effect on telomerase assembly and activity. This assertion is supported by Figure 7c, where it is evident that stabilising the alternate forms of hTR leads to assembly defects, as also mentioned in the manuscript. This conclusion is drawn from the reduced amount of DKC1 and Gar1 pulled down with TERT in the immunoprecipitation (IP) experiments. The mutations were made in hTR but the IPs were performed on TERT. Unlike in the endogenous system, TERT is in vast excess compared to hTR in an overexpression system. Not all overexpressed TERT molecules assemble with hTR. IPs on TERT would only capture hTR that is assembled with TERT, which may not represent the alternative form if these mutations result in assembly defects. Therefore, it remains unclear whether the observed lower enzymatic activity in HEK293T IP samples is a result of "misfolded" hTR, a deficiency in telomerase activity of the assembled enzyme, or an assembly defect that prevents the formation of an active holoenzyme, or a combination of all the above. Therefore, reciprocal experiments with purification on hTR should be performed to examine the effects of the mutations on assembly with TERT and other telomerase holoenzyme components and telomerase activity.
9. The authors mention in the text that the activity observed in the telomerase activity assay for mutant M3 could be attributed

to endogenous wild type hTR present in HEK293T cells. It might be worth considering presenting the levels of activity in HEK293T IP sample without the overexpression of hTR and TERT (which may be undetectable?). This additional information could provide a baseline for endogenous telomerase activity and further support the interpretation of the observed effects in the mutant M3 assay. Alternatively, pulling down on hTR as suggested in the previous point may also cover this point.

10. In Figure 7b, the authors display the activity of telomerase samples derived from HEK293T and HeLa cells. However, there is a discrepancy in the x-axes between the top and lower panels. The top panel's x-axis shows the concentration of telomerase, presumably referring to protein concentration in the IP sample, while the x-axis in the lower panel represents the volume of telomerase used. If this discrepancy is intentional, providing a clear explanation in the figure legend would help readers better understand the rationale behind the choice of axes. Also it is likely that not all overexpressed TERT is assembled with hTR. It would be helpful to describe how telomerase concentration is determined?

11. In Figure 7c, the numbers 5 and 2.5 on top of the gel are not explained. Assuming they refer to loading amounts (μg of total protein measured in HEK293T IP samples), it would be beneficial to either label this information directly on the figure or include it in the legend for clarity. Additionally, providing graphical quantification of DKC1 and Gar1 signals could enhance the interpretation of the results.

12. In Figure 7d, the authors present Northern blot results for hTR in extracts and IP fractions. There are many additional bands for both the WT and the mutants in the IP fractions. Also in the extract lanes, it seems that only the extract from the M2 mutant resembles the wild type, while the mutants M1 and M3 exhibit a notable and distinct band above 527nt. It would be valuable for the authors to clarify whether these bands correspond to an immature form of hTR. Additional labels or descriptions of these differences in the legends would be very helpful for readers to understand the observed variations in the Northern blot results.

13. In the "hTR Northern Blots" section of the methods, it would be advantageous for the authors to include more details about the preparation of the DNA probe used in this study. Providing additional information about the probe preparation, such as the sequence, and any specific conditions used, will enhance the reproducibility of the results presented.

Minor comments/suggestions

14. Suggestion: Consider including a distribution of canonical and alternate forms for the over-expressed wild type of hTR in panel c of Figure 6. This data could be moved from Figure S5, facilitating a more direct comparison between the wild type and mutants. This adjustment might enhance the clarity of the comparative analysis between different hTR variants.

15. Figure 1a is cited in the second paragraph of the introduction following the statement that co-assembly necessitates hTR, TERT, and several additional proteins to form a functional RNP complex. However, Figure 1a displays a cartoon representation of human TERT. It might be more fitting to include a cartoon representation of telomerase composition or simply refer to specific studies.

16. In Figure 6, the M1 mutant is not labeled, while G270C is mentioned in the text. Authors should include the label for the M1 mutant in the figure, like the M2 and M3 mutants.

Figure 1

17. In the cartoon of Figure 1b, it could be clearer if the authors changed the colors of CR4/5 and scaRNA to match the boxes used to highlight these regions (green and orange, respectively). This color consistency would help readers easily associate specific elements in the cartoon with their corresponding highlighted regions.

18. At the end of the figure legend, authors forgot to remove a note they created for themselves. I think it would be appropriate to indicate PDB IDs and reference the studies that the structures originate from.

Figure 2

19. In Figures 2a, 3a, and 4a, the x-axes are not labeled. For consistency, axes should be labelled with nucleotide positions, similar to what was done in Figure 6c.

Figure 4

20. In Figure 4, the font style and size of the x and y axes are not consistent with Figure 2 and Figure 3. It would be advisable to maintain a consistent font style and size across all figures.

21. In the second paragraph of the section "Ensemble deconvolution reveals alternative conformations of tPK and CR4/5 domains," authors highlight relative abundances of hTR tPK as $\sim 80\%$ and $\sim 20\%$ (82% and 18% as in Figure 4a). For consistency, it would be helpful to ensure that the values mentioned in the text match those presented in the corresponding figures. Same applies to relative abundances of CR4/5 presented in Figure 5.

22. Throughout the manuscript, authors use ScaRNA. scaRNA is more commonly used.

23. In various instances throughout the manuscript, figure panels are referred to using lower case letters (e.g., Figure 1c), while in the actual figures, panels are labeled with upper case letters (A, B, C, etc.).

Version 1:

Reviewer comments:

Reviewer #1

(Remarks to the Author)

I am glad to see that the authors took into account most of my comments and suggestions. I still have a couple of points I'd like to raise:

- Concerning the experiments in BJ fibroblasts, HeLa and HeLa+hTERT, the correlations in Fig S6 suggest that some differences do exist between these cell lines, possibly as a consequence of the lack of hTERT. However, it is hard to determine if this represents a true biological difference or, rather, the error associated with the deconvolution analysis. It would be important to show also the inter-replicate correlations across 2X BJ replicates, 2X HeLa replicates, and 2X HeLa+hTERT replicates to defined the "expected" correlation between two reconstructions by DREEM. If these are higher as compared to the inter-cell correlations, this would support the notion that there are indeed differences. If that would be the case, it would be good to at least speculate about the reasons. For instance: is it possible that one or more additional minor conformations might exist in BJ cells? What would happen if the authors would force DREEM to deconvolve 3 or 4 conformations instead of 2? Would the correlation improve? Also, instead of just looking at the MFE structure constrained by the DREEM-deconvolved profiles it would be good to run partition function-based folding and look at the base-pairing probabilities.

- Partly related to the above point, do other software for ensemble deconvolution (e.g., DANCE-MaP, DRACO, DaVinci) agree with DREEM's reconstruction, or do they show something different (or additional conformations)?

- As a comment: While I agree that there is too little sequence variation to perform a covariation analysis, probing the structure of the hTR in mouse cells and showing that the same alternative conformation is also present there would be a stronger proof of the potential functional relevance of this alternative fold. The experiment should be very straightforward to perform and it would enrich the paper.

Reviewer #2

(Remarks to the Author)

The authors have conscientiously and substantially addressed the questions raised by my review, as well as those of the other reviewers. I believe this excellent work is ready for acceptance.

Reviewer #3

(Remarks to the Author)

In the revised manuscript, the authors had done a great job at addressing the comments. They included two additional replicates of the DMS-MapSeq experiments in HeLa cells and DMS-MapSeq experiments in BJ fibroblasts. The experiment of pulldown on hTR we suggested was not performed, which would have further strengthened the conclusion of the assembly defects by the M3 mutant. However, we completely understand that the authors are technically unfamiliar with such experiments. Thus what would be helpful is to raise the point in the discussions.

Reviewer #4

(Remarks to the Author)

Point-by-point Response to Reviewer Comments

We would like to thank all of the reviewers for their helpful comments and suggestions on our manuscript. We appreciate Reviewer 1 stating that our work is '*... potentially interesting and of high relevance*'. We also thank Reviewer 2 for expressing that '*The experiments are technically challenging due in part to the low abundance of telomerase even in cancer (HeLa) cells, and this is the first example of the modern probing techniques and ensemble deconvolution analysis being applied to telomerase RNA.*' Lastly, we appreciate Reviewer 3 pointing out that '*The main novelty of the manuscript is that the authors investigated the alternative conformations of the t/PK and CR4/5 domains of hTR endogenously.*'

Using the reviewer feedback as a guide, we have substantially revised the paper to address many of the concerns that were raised. In addition, we have added several of the suggested new experiments and analyses. We feel the manuscript is now much stronger than the originally submitted version. Below, we describe each of the changes to the manuscript that were made to address the reviewer comments.

Reviewer #1 (Remarks to the Author):

In this manuscript Forino and colleagues analyze the structure of the Telomerase RNA component in human cells, by DMS-MaPseq coupled to DREEM ensemble deconvolution. The story is potentially interesting and of high relevance. However, there is a number of points that the authors need to address to support their claims:

1. - "Our DMS reactivities of the t/PK domain (Fig. 2a and 2b) are in close accord with prior chemical probing studies". Where is this shown or quantified?

- "We detected DMS reactivity in the nucleotides comprising the P6.1 stem (A302, A304, C311, C313), a finding consistent with previous chemical probing studies". As per my previous comment.

We have elected to remove these statements from the manuscript for clarity. Our comparison to previous work in the prior submission was intended to qualitatively place our current results into the context of previously published studies that employed chemical probing techniques to investigate telomerase RNA structure and function.

2. - "DMS reactivity of the CR4/5 domain was consistent with a three-way junction architecture". This should also be quantified. Visual inspection of the data is not sufficient. You can use an AUROC for example.

Thank you to the reviewer for suggesting a more in depth use of AUROC analysis in our study. In the prior version, we introduced AUROC analysis late in the results section. In response to this comment and several of the other reviewer comments we have now added a detailed description of the AUROC analysis, highlighting its utility in evaluating the quality of DMS MaPseq data-driven RNA secondary structure models. We have added a new main text figure (Fig. 4) that shows results of AUROC analysis that evaluates the agreement of our population average reactivity data with widely reported (Chen et al. 2000) secondary structure models of both the template/pseudoknot and CR4/5 domains. This figure also highlights the reproducibility of AUROC values across independent experimental replicates.

In addition, we have included a new supplementary figure (Fig. S2) that shows AUROC analysis of the RNA structure models that emerge from our DREEM deconvolution experiments for both the t/PK and CR4/5 domains. This analysis highlights the scatter in the AUROC values of the

minor (alternative) conformation of the t/PK domain, which suggests that the DREEM method is not able to fully deconvolute the heterogeneity across the t/PK domain. This result supports our decision to focus our subsequent experiments on the more robust DREEM deconvolution results on the CR4/5 domain.

3. - "The results of the deconvolution analysis were reproducible across multiple biological replicates". Not quite true. If I look at Supplementary Fig. S1, it looks like the deconvoluted profiles for the alternative cluster do not well correlate between replicates. I suspect this might be due to the low abundance of this conformer. What was the sequencing depth used to perform this analysis? Would increasing the coverage help? I would also suggest to make another replicate at least.

The reviewer correctly points out that correlation plots for the DREEM deconvoluted alternative model of the t/PK domain (Fig. S1b) are not well correlated. This lack of correlation across replicates in the reactivity pattern for the t/PK domain is in contrast to the CR4/5 domain (Fig. S1). This important observation is another piece of data that we use to highlight the challenges of using DREEM to analyze the t/PK domain compared with the CR4/5 domain.

We have now added two additional biological replicates for both the t/PK and CR4/5 DMS MaPseq experiments performed in HeLa cells. The reproducibility across replicates for both domains was strong when considering the relative proportions of major and minor populations that our output from the DREEM analysis. We have reworded this section of the results to be more explicit and have included the mean percentage values and standard deviation values across the multiple, independent, biological replicates.

4. - The authors perform their analysis in HeLa cells. To really understand the significance of the identified alternative cluster, the authors should also perform a number of analyses here outlined:

a. transfect the hTERT construct in a cell line that does not naturally express any of the telomerase components

b. analyze cells in which high levels of hTERT are present, such as human embryonic stem cells

c. over-express the protein components of the telomerase complex...

These experiments would help elucidating whether proteins play any role in the proper folding of hTERT.

Thank you to the reviewer for making the excellent suggestion to perform DMS MaPseq analysis and DREEM deconvolution in other cellular contexts. After careful consideration of the comments we have elected to include the following new experiments in our study.

- We have added an experiment in BJ fibroblasts, a cell line that constitutively expresses hTR but does not express the hTERT protein subunit. This experiment (together with several additional new experiments) was intended to determine if hTERT expression levels alters the relative proportions of the major and minor clusters in the hTR CR4/5 domain. As described above we chose to focus on the CR4/5 domain for these experiments given the reproducibility and robustness of the DREEM deconvolution as judged by AUROC analysis. To our surprise, the relative proportions of major and minor

DREEM clusters were in close agreement to our experiments done with HeLa cells (Fig. S5 and S6), suggesting that hTERT binding is not required for hTR folding *in vivo*.

- To further test whether hTERT expression levels can alter the relative proportions of the major and minor clusters in our DREEM deconvolution, we performed DMS MaPseq on HeLa cells that have been transiently transfected with a plasmid that overexpresses hTERT. As observed in the BJ fibroblast experiments, overexpression of hTERT in the HeLa cell background did not significantly alter the relative proportions of the major and minor clusters produced by DREEM deconvolution (Fig. S6 and S7).
- We also included an important point raised by Reviewer 2 asking how the DMS MaPseq and DREEM deconvolution experimental results change when the experiment is performed on biochemically purified telomerase RNP complexes that were assembled in HEK293T cells. For this experiment plasmids expressing hTR and FLAG-tagged hTERT were transiently transfected into HEK293T cells according to previously published methods, to overexpress and assemble active telomerase enzymes. These complexes were purified by immunoprecipitation with the FLAG tag and then subjected to DMS MaPseq and DREEM deconvolution. Interestingly, although DREEM found two distinct reactivity clusters, the resultant CR4/5 secondary structure predictions for both were the same and matched the canonical hTR fold. This result supports our conclusion that the alternate CR4/5 conformation is not compatible with assembly into a functional telomerase RNP complex. Comparison of the subtle differences across the DREEM deconvolution clusters reveal the possibility of different RNP conformations (ie. RNA-protein contacts) within the purified telomerase enzyme. This result is in good agreement with recent cryoEM analysis on similarly purified telomerase RNPs which showed substantial sample heterogeneity.

We appreciate the suggestion of studying hESCs to analyze hTR folding in a naturally telomerase positive cell line. This experiment is a major focus of a current project in the laboratory to follow up on the results reported in the present work and to explore the impacts of disease-linked genetically inherited mutations in hTR on the RNA folding landscape. However, given the nascent stage of this work, we feel work with hESCs and iPSCs is beyond the scope of this paper.

5. - The authors only analyze mutations stabilizing the alternative structure. What about mutations stabilizing the accepted structure? Does that increase telomerase activity? I miss that as a proper experimental control.

Given the relatively small fraction of alternatively hTR in HeLa cells, we felt that making mutations to stabilize the canonical fold may only have a marginal impact on overall telomerase activity. This is why we focus on making the more substantial perturbation to the system by dramatically altering the folding landscape of the CR4/5 domain with the mutations that stabilize the alternative CR4/5 conformation.

6. - Is this alternative conformation truly a kinetic trap, and does it have any functional relevance, or is it just an evolutionary bystander/byproduct? For instance, if performing the same analysis in mouse cells, can the authors identify the same 2 conformations? Does covariation analysis support the alternative structure?

We thank the reviewer for this important question regarding the potential function of the alternate conformation of the CR4/5 domain. We raise the possibility of the alternative fold being

Point-by-point Response to Reviewer Comments

a kinetic trap since this is a well-studied phenomenon in RNA folding and ‘may’ apply to our results. We have performed a basic covariation analysis (see figure below) according to methods described in (Tourasse and Dareuille, Bio-protocol, 2020). As can be seen by the figure, the CR (conserved region) 4/5 of hTR is too well conserved for this analysis to be informative (see all the green highlighted nucleotides). The lack of sequence variation across this region among vertebrates prohibits one to make strong conclusions in favor or against the functional significance of the alternative conformation. Therefore, while we favor the notion that the alternative conformation is misfolded with respect to canonical telomerase function, we do allow for the possibility in our discussion of this fold performing some non-canonical biological function as has been suggested previously for the hTR CR4/5 domain.

Reviewer #2 (Remarks to the Author):

The paper presents chemical probing of human telomerase RNA (hTR) in living cells using dimethyl sulfate mutational profiling with sequencing (DMS-MaPseq) and ensemble deconvolution analysis. Specifically, the hTR template/pseudoknot (t/PK) and three-way junction (CR4/5) domains are separately probed and analyzed. The approach used allows for the analysis of more than one conformation if present. The authors find that both the t/PK and CR4/5 adopt a major conformation that seems to conform to what is expected for the assembled TERT-TR and a minor conformation that could not assemble with TERT. For CR4/5, mutations were designed to test the effects of stabilization of the alternate minor conformation and assays showed that they were detrimental to telomerase assembly and activity. The authors propose that the misfolded CR4/5 is either slowly refolded or degraded, as well as an assembly pathway for the telomerase TERT-TR core. The experiments are technically challenging due in part to the low abundance of telomerase even in cancer (HeLa) cells, and this is the first example of the modern

probing techniques and ensemble deconvolution analysis being applied to telomerase RNA. It is a bit disappointing that only the CR4/5 is analyzed in detail and not t/PK, especially as a model for assembly is presented in the last figure.

We thank the reviewer for providing the detailed comments on our initially submitted manuscript. As described above, and in greater detail in the revised manuscript, we elected to focus on the CR4/5 domain results for the latter part of the study due to limitations of deconvolution of the t/PK domain. However, to our knowledge we provide the first direct evidence for folding heterogeneity in the hTR t/PK domain in living cells and provide a possible alternative RNA structural model based upon our DMS MaPseq and DREEM deconvolution results. We have revised the model figure to include our observation of folding heterogeneity in the t/PK domain.

Major:

1. It is not clear to me how/whether the authors distinguish free hTR from hTR assembled with TERT. According to the cited Xi and Cech (2014) paper, HeLa cells have ~240 telomerase/cell, ~1140 hTR, and ~500 TERT. For hTR, that would mean ~21% of the hTR is bound and 79% is free. In the analyzed data, the relative abundance of canonically folded and alternatively folded is 80% and 20% for t/PK and 85% and 15% for CR4/5, respectively. So if the numbers in the Xi and Cech paper are correct, these percentages cannot be for TERT bound vs free, correct?

Thank you to the reviewer for making this important comment. It is correct that the telomerase subunit stoichiometry measured by Xi and Cech (2014) is not consistent with the interpretation that the major and minor clusters found in the present study are due to TERT bound versus free hTR. This conclusion is further substantiated by new experiments added in revision that show the relative proportions of major and minor DREEM clusters do not change in BJ fibroblasts (cells that express hTR but not hTERT), nor do they change in response to ectopic over expression of hTERT in HeLa cells. This is an important finding of our study, which suggests that cellular factors other than hTERT may contribute to hTR folding during telomerase biogenesis. This point is described in the revised manuscript.

Also, wouldn't hTR bound to TERT have different reactivity from free hTR even if the fold is 'canonical'?

The reviewer raises an important caveat when considering how to interpret DMS MaPseq reactivity data. Certainly one might expect that hTR bound to TERT could react with DMS differently if an exposed base pairing face of an adenine or cytosine base becomes buried upon RNP complex formation. However, our data do not show any obvious signs of such "footprinting." The very small DMS probe appears to recognize RNA structure even if a protein is bound. In support of this conclusion, published data from DMS MaPseq studies performed on HIV and the yeast ribosome indicate that exposed bases react with the small DMS chemical probe, even when part of an assembled RNP (Rouskin et al. *Nature* 2014; Tomezsko et al. *Nature* 2020).

We have now included a DMS MaPseq experiment on biochemically purified telomerase RNPs (Fig. 9). The complexes are purified using FLAG-tagged hTERT and the approach should enrich for hTR that is assembled into RNP complex and deplete contributions from free hTR. In this experiment, we still observe sufficient heterogeneity in the DMS reactivity signal to drive DREEM deconvolution into two distinct clusters. However, the data-driven models that arise from these DREEM clusters are the same and agree with the canonical CR4/5 fold. This

observation suggests that DMS MaPseq can detect subtle differences in RNP complex structure, particularly in unpaired regions of hTR that make protein contacts.

In that case, wouldn't one expect 3 populations, free hTR 'canonical', free hTR alternately folded, and bound TR?

See response above.

2. To help answer the question above, have the authors tried DMS/SHAPE+deconvolution on in vitro purified telomerase (using the FLAG-tagged hTERT purified telomerase RNA from the FLAG-hTERT and WT hTR transfected cells)? I think this control is essential, for both CR4/5 and t/PK. Also map these results on the highest resolution human telomerase structures and compare to results for the in vivo probed telomerase.

We have now added an analysis of the hTR CR4/5 domain using DMS MaPseq and DREEM deconvolution on biochemically purified telomerase RNPs. As described above, our DREEM results on the t/PK domain suggest that there is more heterogeneity than can be reproducibly captured by our approach. For this reason, we focus on the CR4/5 domain. Importantly, the DREEM deconvoluted DMS reactivity patterns derived from the purified telomerase support a canonical CR4/5 fold, whereas the alternative fold is not observed. This result supports our conclusion that the alternative fold is not competent to assemble into the telomerase RNP.

3. In the introduction, the authors mention "Overexpression of telomerase has also historically been used to mitigate problems arising from its scarcity but has been shown to bypass the endogenous RNP assembly pathway". For the CR4/5 variants, the authors overexpressed hTR and hTERT and transiently transfected to analyze their effects on structure in vivo. They did do the control probing for WT hTR (Fig S5), and say in the main text they found an increase in alternatively folded (~30% vs ~13%). However the two DMS profiles they show in the SI have ratios of 79% vs 21% and 67% vs 33%, which does not average to 30% for the 'misfolded' conformation.

We thank the reviewer for pointing out this issue in the original manuscript. We have now carefully revised the paper to correctly average and indicate standard deviations for the relative abundances of the different populations observed by DREEM deconvolution across independent biological replicates.

More importantly, the profiles look different from the results on endogenous hTR and to some extent to each other (compare Fig 5 and Fig S5). Please explain.

The reviewer is correct that there is some variation in the DMS reactivity profiles, particularly for the experiments where transiently transfected hTR is chemically probed. As noted in the manuscript, this variation could be due to differences between experiments in the level of hTR expression and the impact of overexpression on other cellular factors that may aid hTR folding.

4. It would be useful to also see the t/PK results for overexpression vs endogenous vs in vitro purified telomerase.

As described above, based upon our initial analysis of the reproducibility of the DMS reactivity and DREEM deconvolution in the t/PK domain, we elected to focus our analysis on the CR4/5 domain for this section of the study.

5. Figure 8 “Model of telomerase RNP assembly with respect to hTR folding” and discussion completely ignores assembly with t/PK and only shows t/PK in ‘canonical’ conformation. Does CR4/5 bind first and then t/PK or vice versa? If one is showing assembly, why aren’t the H/ACA proteins shown binding first? I suggest deleting or changing this figure.

We have revised the model figure to incorporate our observation of heterogeneous folding of the t/PK domain as well as the CR4/5 domain. In addition, since H/ACA protein binding is thought to be an early step in telomerase assembly, we have now included them in the model for clarity.

6. In all the figures that show reactivity, the authors need to clearly distinguish between no detection and no/low reactivity. Only Cs and As seem to show any reactivity in these experiments (although DMS can be used to probe U and G).

The reviewer is correct that modified U and G bases were not analyzed in this study. In all figures, these bases are indicated in white, whereas no/low reactivity (<0.1) of A and C bases is shown in grey. In Zubradt et al. Nature Methods (2016), a comprehensive analysis of the ability of TIGRT to produce mismatches at methylated C and A bases, and very rarely at U and G bases, is shown. Therefore, we focus our analysis on the robust properties of the reverse transcriptase employed in our study.

7. Discussion “hTR structure is known to adopt a phylogenetically conserved secondary structure (ref 10) and therefore would not be predicted to adopt long-lived alternative conformations in vivo.” I think I understand what the authors mean to say here, but it is not clear why an alternative structure could not form stably in absence of TERT (as proposed in the references cited in discussion, second page second full paragraph). Isn’t there some evidence that hTR co-folds with TERT?

To our knowledge, there is no direct evidence that binding of TERT in vivo results in a significant rearrangement of hTR base pairing configuration. There is evidence for protein-induced conformational changes in telomerase RNAs from other systems, such as Tetrahymena (ie. Stone et al. Nature 2009) – however, these changes were not shown to correspond to significant reorganization of the RNA secondary structure. The point we are making here is that one would not expect long-lived hTR conformations with substantially reorganized secondary structures to persist in vivo. We have edited this sentence to make this point clearer.

8. The authors do a nice job of referencing previous literature in the introduction, with a couple of notable exceptions. Specifically, the high-resolution human telomerase cryo-EM structures listed do not include Wan, et al Cell Res (2021) 10.1038/s41422-021-00586-7.

We have added the requested citation to the introduction.

9. Discussion, last paragraph: “These same techniques could be used to explore the populations of other functional noncoding RNAs in cells.” Haven’t similar techniques been used already for lncRNAs, e.g. for 7SK RNA? 10.1016/j.molcel.2022.02.009

Reference to this study has now been added to the discussion.

10. For Figure 1, the last line reads “GIVE EITHER PDB NUMBERS OR CITE REF FOR c AND d.” Please correct.

Thank you for catching this error, we have now included the appropriate PDB number.

11. METHODS: The methods needs to be carefully proofread. In “Cell culture and DMS modification” 100 units/mL of penicillin was used but units for streptomycin are 100 mg/m (?). Should 10 mL of ice cold DMS quench solution be 1 ml? For the Western blots and Telomerase activity assays, in several places there is m instead of (micro), incubation is at 80 °C (!). In hTR northern blots the degree sign before C is missing and “counts” should be counts per minute?

We have carefully proofread and expanded the methods section and corrected each of the errors. Thank you for pointing these out.

12. Please put the % of each population on Figure 4 and 5. Please provide uncertainties for the multiple replicates of the chemical probing.

We have included the percentages for each population shown in each DREEM deconvolution experiment. Uncertainties across independent biological replicates are now cited in the main text.

13. Fig. 6 G270 should be G270C (its wrong 3 times). Also, please include results of probing on WT in this figure for comparison.

Thank you for catching this error, we have now corrected it in the figure. We have also added the results of the overexpressed WT hTR for comparison to each of the mutant CR4/5 hTR constructs. As described in the main text, we observed an elevated proportion of the alternative conformation for the overexpressed WT hTR when compared with endogenous hTR levels, suggesting that overexpression may slightly perturb the folding equilibrium. In addition, we note that the reactivity pattern for the overexpressed WT hTR appears more heterogeneous than the pattern for the endogenous hTR or any of the CR4/5 mutants designed to stabilize the alternative fold.

Reviewer #3 (Remarks to the Author):

In this manuscript, Forino et al., used dimethyl sulfate mutational profiling combined with sequencing (DMS-MapSeq) to probe the structure of endogenous hTR in HeLa cells. They used primers specific for the t/PK and CR4/5 domains to probe DMS reactivities for these domains of hTR. While the reactivities of most nucleotides agree with the predicted secondary structures (SS) of hTR and cryoEM structures of human telomerase, there are a few nucleotides whose reactivities do not fit the predicted SS. The authors then used the DREEM algorithm to deconvolute their data into major and minor clusters and then used the clusters to guide thermodynamic predictions of different conformations of the t/PK and CR4/5 domains. The major and minor clusters were found to correspond to the canonical conformation and alternative conformations for each of t/PK and CR4/5, respectively. They chose to perform further mutagenesis experiments on the CR4/5, which showed better results from their experiments and modelling. They identified three

positions in CR4/5 which were predicted to disrupt base-pairing of the canonical conformation of the RNA domain and/or stabilise the alternative conformation. They overexpressed hTR single, double or triple mutants at these positions with TERT in HeLa and performed DMS-MapSeq and DREEM deconvolution. They found that the single and double mutants showed a small increase in the abundance of the alternative conformation of CR4/5 compared to cells overexpressed with WT hTR. The triple mutant showed a large increase in the alternative conformation. They also performed immunoprecipitation on TERT and showed that the single and double mutants only showed a small reduction in the total telomerase activity and the triple mutant retained very little telomerase activity. Similarly, the amounts of dyskerin and GAR1 in the mutants were reduced moderately in the single and double mutants and drastically for the triple mutant.

Some previous studies have described alternative conformations of t/PK and CR4/5 but in an in vitro context. The main novelty of the manuscript is that the authors investigated the alternative conformations of the t/PK and CR4/5 domains of hTR endogenously. The effect of the triple mutant identified here is very intriguing, which would form the basis for further investigation. However, with the presented data, it is still not quite clear how exactly the alternative conformations directly contribute to telomerase assembly. Furthermore, due to the lack of detailed explanations of the methods employed, I find the manuscript a bit difficult to read and understand how the results are generated and what the pitfalls are. There are a number of specific points that the authors should consider:

1. I appreciate that the authors have included a short description of DMS-MaPseq in the introduction. It would be beneficial to provide more detailed explanations of how the methods (e.g. DMS-MaPseq, DREEM, AUROC) work and how they were specifically implemented in this study. This additional detail will enable non-specialist readers to better grasp the main points the authors aim to convey. More specific suggestions will be in some of the points below.

Thank you for this comment. Since both DMS-MaPseq and DREEM have been described in detail in previous publications we tried to give the reader the necessary knowledge to appreciate how these methods work and how we are specifically applying them for our study of telomerase RNA folding. In the case of the AUROC analysis, we have now added a new detailed description of how the analysis works and why it is a useful approach for evaluating the quality of data-driven models of RNA secondary structure. We have also expanded the methods section to be a bit more comprehensive for all of the experimental approaches employed in our study so that the interested reader can find the detailed explanations and appropriate references there.

2. In Figures 4 and 5, the authors compare the canonical and alternative structures of hTR regions. In Figure 6, they further characterise these regions by concentrating on three residues that stabilise the alternate conformation of hTR. I noticed a few other distinctions between the canonical and alternate forms of hTR, such as t/PK (194, 197-180, region around 150-170) or CR4/5 (240-242, 313-318). Is there any specific reason why these differences are not discussed or pursued with a more detailed analysis?

The reviewer is correct that there are several reactivity differences between the canonical and alternative forms of both the t/PK and CR4/5 domains. We specifically focused our mutational analysis on the selected mutations because in silico structure prediction of these sequences suggested they would have the simultaneous ability to destabilize the canonical fold while

stabilizing the alternative fold. Our goal was to find strategically chosen mutations that would permit us to tune the folding landscape of the CR4/5 domain toward the alternative fold so that we can investigate the functional impacts of this stabilizing effect. Our DMS MaPseq experiments confirm that the selected mutations do in fact bias the folding towards the alternate fold as intended. Since these RNA mutations worked in the manner that we hypothesized we did not explore other possible analyses of reactivity differences between the canonical and alternate hTR conformations.

3. In Figures 4 and 5, there are many residues which are involved in base-pairing but exhibit considerable DMS-reactivities (4b: 62, 66, 74, 75, 79, 126-129, 138, 148, 190, 196; 4c: 35, 61, 62, 66, 74, 75, 79, 104, 123, 126-128, 131, 138, 148, 149, 164, 169, 170, 190, 196; 5b: 255, 295, 302, 304, 313, 320, 321; 5c: 240, 252, 266, 267, 285, 301, 302, 320, 321, 328, 330). How are these reactivities accounted for?

It is not unusual for there to be reactivity at nucleotides that are modeled to be structured as the reviewer points out. Some of this reactivity signal may be due to transient 'breathing' of the RNA structure during the period of DMS exposure. In other cases, these reactive nucleotides may represent some subset of molecules that are in some alternatively folded state but likely represents a minority of the RNA folding ensemble. Ultimately, all of the reactivity data are included in the data-driven modeling as a pseudo energy term, such that strongly reactive nucleotides incur a large penalty in calculating the stability of a given RNA structure model. It is also important to note that all of the reactivities across the RNA model are considered as part of the AUROC analysis. Thus, if there was a high level of reactivity in regions predicted to be base-paired then this would result in a lower AUROC value.

4. How do DMS reactivities of nucleotides change in respond to binding to protein partners (such as TERT or histones)? I am not familiar with the field so it would be good for the authors to take this into account in their results/discussions.

Please see response to Reviewer 2, Major comment 1.

5. The DREEM deconvolution yields only two conformations of each of the t/PK and CR4/5 domains. I would expect that endogenously hTR would have more conformations than just two. How likely is it that more conformations may exist that could not be captured by the deconvolution method?

Another excellent point, as it is quite likely that there are more than two conformations. The DREEM deconvolution method sets a conservative threshold for fitting more than two conformations to avoid possible artifacts and overfitting of the data. This is where the AUROC analysis is very important, as scatter in the AUROC value across experimental replicates suggests the presence of more heterogeneity than DREEM can reliably deconvolute.

6. The normalisation procedure for DMS reactivity in the reads mentioned in Figures 2, 3, 4, 5, and 6 is not explicitly outlined in the methods or the text. It would be helpful for a more comprehensive understanding of the study to include details about how DMS reactivity was normalised in these figures.

Thank you for pointing out this omission. We have expanded our discussion in the methods section of how the data were analyzed and normalized to generate the DMS reactivity values plotted throughout the manuscript.

7. They also calculated the area under the receiver operating characteristic curve (AUROC) to evaluate the quality of their models, which are shown in Fig. S4. I do not understand what this Figure shows and how the overall values are derived from this figure. What do the two colors in the yellow bars represent? How are the DMS modification levels shown in the blue bar normalised? It is also intriguing that the t/PK canonical form has the lowest confidence level. Is there an explanation for this?

In the revised version of the paper we have completely rewritten the description of the AUROC analysis and made a variety of new figures to show how the AUROC curves are plotted. The reviewer makes the great observation that the t/PK AUROC values are consistently lower and have a higher degree of scatter across biological replicates. This observation likely reflects challenges in RNA structure folding software to predict RNA pseudoknots – which is a well-known issue in the RNA structure prediction field. The lower AUROC values also indicate that the models are less well supported by the data, which may be a consequence of the t/PK domain exhibiting a high degree of heterogeneity than DREEM can reliably deconvolute. These observations motivated us to focus the second half of the manuscript on the more robustly modeled CR4/5 domain conformations.

8. In the abstract, the authors argue that stabilising the alternate forms of CR4/5 has a detrimental effect on telomerase assembly and activity. This assertion is supported by Figure 7c, where it is evident that stabilising the alternate forms of hTR leads to assembly defects, as also mentioned in the manuscript. This conclusion is drawn from the reduced amount of DKC1 and Gar1 pulled down with TERT in the immunoprecipitation (IP) experiments. The mutations were made in hTR but the IPs were performed on TERT. Unlike in the endogenous system, TERT is in vast excess compared to hTR in an overexpression system. Not all overexpressed TERT molecules assemble with hTR. IPs on TERT would only capture hTR that is assembled with TERT, which may not represent the alternative form if these mutations result in assembly defects. Therefore, it remains unclear whether the observed lower enzymatic activity in HEK293T IP samples is a result of "misfolded" hTR, a deficiency in telomerase activity of the assembled enzyme, or an assembly defect that prevents the formation of an active holoenzyme, or a combination of all the above.

The reviewer is correct to point out that our observation of reduced pull down of DKC1 and Gar1 in the context of the hTR mutants suggests these mutations exhibit an RNP assembly defect. Since the experimental design is to pull down the FLAG-tagged hTERT, we conclude that less hTR, and therefore less DKC1 and Gar1, are coassembled with hTERT in the presence of the mutant hTR constructs. This result is most pronounced for the M3 mutant, which shows the most dramatic shift in folding towards the alternative CR4/5 conformation, and also the most dramatic loss of catalytic activity when assayed in vitro. It is correct that our results cannot eliminate the possibility that some of the alternative conformation of hTR can assemble with hTERT but result in knocked down activity. However, given our structural knowledge of protein-RNA contacts made with the three-way junction motif within the CR4/5 domain, the most likely explanation for our results is that the alternative fold cannot efficiently assemble with hTERT.

Therefore, reciprocal experiments with purification on hTR should be performed to examine the effects of the mutations on assembly with TERT and other telomerase holoenzyme components and telomerase activity.

As the reviewer likely knows, the reciprocal purification experiments that are suggested are not trivial and we feel that our IP experiments in their current form, looking at Western blots of

DKC1 and Gar1, Northern blots of hTR pulldown during FLAG IP, and direct primer extension assays provide a clear view of the deleterious impacts of the M3 hTR mutation that stabilizes the alternative conformation of the CR4/5 domain.

9. The authors mention in the text that the activity observed in the telomerase activity assay for mutant M3 could be attributed to endogenous wild type hTR present in HEK293T cells. It might be worth considering presenting the levels of activity in HEK293T IP sample without the overexpression of hTR and TERT (which may be undetectable?). This additional information could provide a baseline for endogenous telomerase activity and further support the interpretation of the observed effects in the mutant M3 assay. Alternatively, pulling down on hTR as suggested in the previous point may also cover this point.

Thank you for this suggestion. However, since about 10% of the M3 mutant RNA retains a DMS reactivity pattern that is consistent with the canonical fold, it is also possible that the residual activity is due to properly folded M3 hTR. We have revised the text to include this additional possibility.

10. In Figure 7b, the authors display the activity of telomerase samples derived from HEK293T and HeLa cells. However, there is a discrepancy in the x-axes between the top and lower panels. The top panel's x-axis shows the concentration of telomerase, presumably referring to protein concentration in the IP sample, while the x-axis in the lower panel represents the volume of telomerase used. If this discrepancy is intentional, providing a clear explanation in the figure legend would help readers better understand the rationale behind the choice of axes. Also it is likely that not all overexpressed TERT is assembled with hTR. It would be helpful to describe how telomerase concentration is determined?

We have corrected these plots so that they both show telomerase concentration, rather than volume. Thank you for pointing out this discrepancy. In addition, we have added a line in the methods describing how the telomerase concentration was determined.

11. In Figure 7c, the numbers 5 and 2.5 on top of the gel are not explained. Assuming they refer to loading amounts (μg of total protein measured in HEK293T IP samples), it would be beneficial to either label this information directly on the figure or include it in the legend for clarity. Additionally, providing graphical quantification of DKC1 and Gar1 signals could enhance the interpretation of the results.

The numbers in the figure refer to μL amount of sample loaded to ensure that the results are consistent across different loading amounts. We have labeled the figure accordingly. We have kept with the numbers below the Western Blots for this section of the figure since the figure already has a large number of panels.

12. In Figure 7d, the authors present Northern blot results for hTR in extracts and IP fractions. There are many additional bands for both the WT and the mutants in the IP fractions. Also in the extract lanes, it seems that only the extract from the M2 mutant resembles the wild type, while the mutants M1 and M3 exhibit a notable and distinct band above 527nt. It would be valuable for the authors to clarify whether these bands

correspond to an immature form of hTR. Additional labels or descriptions of these differences in the legends would be very helpful for readers to understand the observed variations in the Northern blot results.

In the revised version we now include substantially optimized Northern blots which show far fewer nonspecific bands. While a low level of shorter products persists, we do not know whether this is nonspecific probe binding or some hTR degradation products. We also characterized the behavior of full-length hTR on our blots, showing that heat treatment followed by snap-cooling results in full-length hTR running as a doublet as has been seen in previous publications. We have also included probes to U1 and U2 snRNAs for normalization procedures as described in the revised paper.

13. In the "hTR Northern Blots" section of the methods, it would be advantageous for the authors to include more details about the preparation of the DNA probe used in this study. Providing additional information about the probe preparation, such as the sequence, and any specific conditions used, will enhance the reproducibility of the results presented.

We have now expanded discussion of the Northern Blot procedure and included the DNA probe sequences in Supplementary Table S1.

Minor comments/suggestions

14. Suggestion: Consider including a distribution of canonical and alternate forms for the over-expressed wild type of hTR in panel c of Figure 6. This data could be moved from Figure S5, facilitating a more direct comparison between the wild type and mutants. This adjustment might enhance the clarity of the comparative analysis between different hTR variants.

Thank you for this suggestion, we have now moved the data for the WT hTR overexpression experiment into the main text figure alongside each of the mutant hTR constructs (Fig. 7 in the new revision).

15. Figure 1a is cited in the second paragraph of the introduction following the statement that co-assembly necessitates hTR, TERT, and several additional proteins to form a functional RNP complex. However, Figure 1a displays a cartoon representation of human TERT. It might be more fitting to include a cartoon representation of telomerase composition or simply refer to specific studies.

In Figure 1 we elect to focus on the catalytic core of the telomerase RNP for clarity. However, we did add references to recent telomerase structural biology reviews which provide detailed descriptions of telomerase RNP composition.

16. In Figure 6, the M1 mutant is not labeled, while G270C is mentioned in the text. Authors should include the label for the M1 mutant in the figure, like the M2 and M3 mutants.

This omission has been corrected, thank you.

Figure 1

17. In the cartoon of Figure 1b, it could be clearer if the authors changed the colors of CR4/5 and scaRNA to match the boxes used to highlight these regions (green and orange, respectively). This color consistency would help readers easily associate specific elements in the cartoon with their corresponding highlighted regions.

The colors in the figures have been made to match the boxes used to highlight the regions. We note that the ScaRNA domain is not shown in the structure figures shown in panels c and d.

18. At the end of the figure legend, authors forgot to remove a note they created for themselves. I think it would be appropriate to indicate PDB IDs and reference the studies that the structures originate from.

Thank you, we have corrected this mistake and have added the PDB IDs. The appropriate references are included in the body of the text.

Figure 2

19. In Figures 2a, 3a, and 4a, the x-axes are not labeled. For consistency, axes should be labelled with nucleotide positions, similar to what was done in Figure 6c.

Thank you for pointing out this issue – we have revised all pertinent figures as suggested.

Figure 4

20. In Figure 4, the font style and size of the x and y axes are not consistent with Figure 2 and Figure 3. It would be advisable to maintain a consistent font style and size across all figures.

We have gone through all figures to match font styles and sizes.

21. In the second paragraph of the section "Ensemble deconvolution reveals alternative conformations of t/PK and CR4/5 domains," authors highlight relative abundances of hTR t/PK as ~80% and ~20% (82% and 18% as in Figure 4a). For consistency, it would be helpful to ensure that the values mentioned in the text match those presented in the corresponding figures. Same applies to relative abundances of CR4/5 presented in Figure 5.

In the revised manuscript, the cited values are described in the text as averages with standard deviations from multiple independent biological replicates. In the case of the figures, the values shown are for the specific replicate used in the figure.

22. Throughout the manuscript, authors use ScaRNA. scaRNA is more commonly used.

Thank you for this note, we have corrected the text.

23. In various instances throughout the manuscript, figure panels are referred to using lower case letters (e.g., Figure 1c), while in the actual figures, panels are labeled with upper case letters (A, B, C, etc.).

Point-by-point Response to Reviewer Comments

All figure panels and references to figures in the text are now in Nature format using lower case letters.

Reviewer #1 (Remarks to the Author):

I am glad to see that the authors took into account most of my comments and suggestions. I still have a couple of points I'd like to raise:

We thank the reviewer for the excellent feedback in the first round of reviews.

- Concerning the experiments in BJ fibroblasts, HeLa and HeLa+hTERT, the correlations in Fig S6 suggest that some differences do exist between these cell lines, possibly as a consequence of the lack of hTERT. However, it is hard to determine if this represents a true biological difference or, rather, the error associated with the deconvolution analysis. It would be important to show also the inter-replicate correlations across 2X BJ replicates, 2X HeLa replicates, and 2X HeLa+hTERT replicates to defined the "expected" correlation between two reconstructions by DREEM. If these are higher as compared to the inter-cell correlations, this would support the notion that there are indeed differences. If that would be the case, it would be good to at least speculate about the reasons. For instance: is it possible that one or more additional minor conformations might exist in BJ cells? What would happen if the authors would force DREEM to deconvolve 3 or 4 conformations instead of 2? Would the correlation improve? Also, instead of just looking at the MFE structure constrained by the DREEM-deconvolved profiles it would be good to run partition function-based folding and look at the base-pairing probabilities.

We agree that the differences in deconvolution results between cell types does suggest that there may be some cell type specific differences in the hTR folding ensemble. We have now included an additional correlation analysis of the BJ fibroblast experiment (TERT negative) and show in Supplementary Figure 6 that inter-replicate correlations are very strong for this cell type, similar to our previously reported results with HeLa cells. Given that we show in both HeLa cells and BJ fibroblasts that the inter-replicate correlations are very high, we opted not to repeat the analysis for the HeLa - hTERT experiment which may suffer from transfection efficiency induced variability. Nevertheless, to address the point of cell type specific variation, we have also added a figure in Supplementary Figure 8 which directly shows the variability across experiments performed in the different cell types and conditions. We note that the general results of there being a major (canonical) and minor (alternative) hTR conformation is highly reproducible across all conditions, despite there being some variability at the individual nucleotide reactivity level as the reviewer correctly points out.

We agree that while DREEM determines there to be two clusters, this simply means there are at least two populations and there could in principle be more. However, the DREEM software has very conservative overfitting criteria. Thus, we agree that our results place a lower bound on the degree of heterogeneity in the hTR ensemble, but we prefer to take this parsimonious approach rather than overinterpreting the data. We have discussed this point in the paper.

We also agree that looking at the partition function-based folding to get base-pairing probabilities could be an interesting analysis to perform. However, given that we are analyzing the top three MFE structures for each population described in this study, and this analysis approach has precedent in the literature, we prefer to keep the data treatment as it is to draw the conclusions in this study. A more detailed analysis of the base pairing probabilities across the heterogeneous hTR folding ensemble could certainly be an interesting direction for future studies.

- Partly related to the above point, do other software for ensemble deconvolution (e.g., DANCE-MaP, DRACO, DaVinci) agree with DREEM's reconstruction, or do they show something different (or additional conformations)?

This is certainly an interesting question - ie. how does DREEM compare with other structure deconvolution algorithm. However, delving into that comparison between different algorithms will be an important study unto itself and we feel this is beyond the scope of the current work which we want to stay focused on hTR folding.

- As a comment: While I agree that there is too little sequence variation to perform a covariation analysis, probing the structure of the hTR in mouse cells and showing that the same alternative conformation is also present there would be a stronger proof of the potential functional relevance of this alternative fold. The experiment should be very straightforward to perform and it would enrich the paper.

Again, we agree that probing in mouse cells would be an interesting avenue to explore; however, our focus in this study is on the human system and since we have studied hTR folding in a variety of human cell contexts, we feel expanding to mouse cells is beyond the scope of this study.

Reviewer #2 (Remarks to the Author):

The authors have conscientiously and substantially addressed the questions raised by my review, as well as those of the other reviewers. I believe this excellent work is ready for acceptance.

Thank you to the reviewer for the very helpful comments in the first round of review.

Reviewer #3 (Remarks to the Author):

In the revised manuscript, the authors had done a great job at addressing the comments. They included two additional replicates of the DMS-MapSeq experiments in HeLa cells and DMS-MapSeq experiments in BJ fibroblasts. The experiment of pulldown on hTR we suggested was not performed, which would have further strengthened the conclusion of the assembly defects by the M3 mutant. However, we completely understand that the authors are technically unfamiliar with such experiments. Thus what would be helpful is to raise the point in the discussions.

Thank you to this reviewer for the previous comments. We note that performing the hTR pull down experiments were not included in the revised manuscript, since, in our opinion, the data presented on the pull down of H/ACA RNP proteins in the hTERT IP experiments demonstrates that hTR is not efficiently assembling into the telomerase RNP in the context of the M3 hTR mutant that favors the alternative conformation. The reviewer is correct that we are not currently set up to perform the experiment using the hTR pulldown, but we can certainly consider this option for future experiments.

Reviewer #4 (Remarks to the Author):
